# Proteogenomic characterization of MiT family translocation renal cell carcinoma

Yuanyuan Qu[1,2,6], Xiaohui Wu [1,6], Aihetaimujiang Anwaier[1,2,6], Jinwen Feng [1,6], Wenhao Xu[1,2,6], Xiaoru Pei[1,6], Yu Zhu[1,2,6], Yang Liu [1], Lin Bai [1], Guojian Yang[1], Xi Tian[1,2], Jiaqi Su[1,2], Guo-Hai Shi[1,2], Da-Long Cao[1,2], Fujiang Xu [3], Yue Wang[1,2], Hua-Lei Gan[2,4], Shujuan Ni[2,4], Meng-Hong Sun[2,4], Jian-Yuan Zhao [5] ✉, Hailiang Zhang [1,2] ✉, Dingwei Ye [1,2] ✉ & Chen Ding [1] ✉

Microphthalmia transcription factor (MiT) family translocation renal cell carcinoma (tRCC) is a rare type of kidney cancer, which is not well characterized. Here we show the comprehensive proteogenomic analysis of tRCC tumors and normal adjacent tissues to elucidate the molecular landscape of this disease. Our study reveals that defective DNA repair plays an important role in tRCC carcinogenesis and progression. Metabolic processes are markedly dysregulated at both the mRNA and protein levels. Proteomic and phosphoproteome data identify mTOR signaling pathway as a potential therapeutic target. Moreover, molecular subtyping and immune infiltration analysis characterize the inter-tumoral heterogeneity of tRCC. Multi-omic integration reveals the dysregulation of cellular processes affected by genomic alterations, including oxidative phosphorylation, autophagy, transcription factor activity, and proteasome function. This study represents a comprehensive proteogenomic analysis of tRCC, providing valuable insights into its biological mechanisms, disease diagnosis, and prognostication.

Microphthalmia transcription factor (MiT) family translocation renal cell carcinoma (tRCC) is a rare renal cancer subtype characterized by chromosomal translocations involving transcription factor E3 (*TFE3*) or EB (*TFEB*) (on chromosomal loci Xp11.2 and 6p21, respectively) genes fusions with various partners[1]. Owing to tRCCs with *TFE3* and *TFEB* gene fusions sharing many clinical, histopathological, and genetic similarities, the 2013 International Society of Urological Pathology (ISUP) Vancouver classification grouped these two subtypes into a single entity, called MiT family tRCC[2]. So far, several gene fusions for *TFE3* have been identified in patients with tRCC, including *ASPSCR1*

(*ASPL*), *PRCC*, *SFPQ* (*PSF*), *NONO*, *CLTC*, *DVL2*, *LUC7L3*, *PARP14*, *MED15*, *KHSRP*, *RBM10*, *ARID1B*, *MATR3*, *FUBP1*, *NEAT1*, *KAT6A*, *GRIPAP1*, and *EWSR1*[3–11]. *ASPSCR1* and *PRCC* were identified as the most common fusion partners of *TFE3*[12–14]. Additionally, several case reports have reported different fusion partners for *TFEB*, such as *MALAT1*, *CLTC*, *ACTB*, *KHDRBS2*, *COL21A1*, *CADM2*, *EWSR1*, and *PPP1R10*[15].

Histologically, tRCC mimics almost all subtypes of renal cell carcinoma (RCC). It can present with papillary architecture and cells with voluminous clear or eosinophilic cytoplasm[12,16]. Some cases even comprise perivascular epithelioid neoplasm-like or melanoma-like

[1]Department of Urology, Fudan University Shanghai Cancer Center, State Key Laboratory of Genetic Engineering, Collaborative Innovation Center for Genetics and Development, School of Life Sciences, Institute of Biomedical Sciences, and Human Phenome Institute, Fudan University, Shanghai 200433, China. [2]Department of Oncology, Shanghai Medical College, Shanghai Genitourinary Cancer Institute, Shanghai 200032, China. [3]Department of Oncology, The Affiliated Hospital of Southwest Medical University, Luzhou 646000, China. [4]Tissue Bank & Department of Pathology, Fudan University Shanghai Cancer Center, Shanghai 200032, China. [5]Institute for Developmental and Regenerative Cardiovascular Medicine, MOE-Shanghai Key Laboratory of Children's Environmental Health, Xinhua Hospital, Shanghai Jiao Tong University School of Medicine, Shanghai 200092, China. [6]These authors contributed equally: Yuanyuan Qu, Xiaohui Wu, Aihetaimujiang Anwaier, Jinwen Feng, Wenhao Xu, Xiaoru Pei, Yu Zhu. ✉e-mail: zhaojy@fudan.edu.cn; zhanghl918@163.com; dwyeli@163.com; chend@fudan.edu.cn

differentiation cells[17], which brings serious challenges in both diagnosis and treatment. Clinically, tRCC classically affects pediatric patients and young adults but the disease has also been observed in older patients in recent years. tRCC accounted for ~20–40%[18,19] of pediatric RCC and 1–4% of adult RCC[20,21]. The incidence of tRCC is controversial due to the difficulty of diagnosis and its diverse subtypes[5]. An analysis of 403 tRCC cases from the literature indicated that the incidence of this disease is slightly higher in females (F:M ratio, 1.6:1)[22]. The prognosis of tRCC remains controversial, ranging from indolent to highly aggressive. Several studies have claimed that adult-onset tRCC is more invasive and aggressive, especially in male patients[4,23–25], but is relatively indolent in pediatric patients[26]. In general, tRCC has a similar prognosis with clear cell RCC (ccRCC)[24,27] but has worse outcomes than papillary RCC[4]. Among all the tRCC types, patients with *ASPSCR1* fusion have the most unfavorable prognosis and lymph node metastasis is more common; however, the correlation between fusion type and prognosis is unclear[28].

Recently, some studies have reported different fusion partners for *TFE3*, but owing to the rarity of such tumors, general understanding of tRCC molecular characteristics and underlying mechanisms has been limited. The detection of copy number variations (CNVs) in 16 tRCC cases demonstrated that the most frequent mutations were 17q gain and 9p deletion, and distinct somatic CNVs were associated with a poor prognosis[29]. However, because previous studies are limited to relatively small cohorts and focused on the detection of genomic alterations, they do not sufficiently explain the biological features of tRCC[30]. Therefore, there is an urgent need for multiomics analyses, especially proteomics and phosphoproteomics analyses.

In this work, we investigate the genomic, transcriptomic, proteomic, and phosphoproteomic characteristics of 86 tRCC samples and correlate these findings with clinicopathological features. We find that mTOR signaling pathway is upregulated in tRCC tumor tissues in both proteome and phosphoproteome levels, which indicates mTOR signaling is a potential therapeutic target. Moreover, our study reveals the impacts on clinical outcomes and molecular features of genomic alterations in tRCC, providing biological insights of tRCC carcinogenesis and development.

## Results

### Molecular Profiling of MiT Family tRCC

We collected a total of 86 MiT family tRCC samples from treatment-naive Chinese patients. A schematic diagram of the experimental design is shown in Fig. 1a. Whole-exome sequencing (WES) was conducted on 84 paired samples to detect any in MiT family tRCC genetic variants. Samples from 2 patients were excluded due to low DNA quality. Label-free proteomic and phosphoproteomic approaches were carried out on 74 tumors and 57 normal adjacent tissues (NATs), and 28 tumors and 21 NATs, respectively. RNA sequencing (RNA-seq) was carried out on 26 tumors and 16 NATs. All omics experiments including WES, transcriptome, proteome, phosphoproteome were conducted on the same patients and control.

This cohort was comprised by 33.7% (*n* = 29) males and 66.3% (*n* = 57) females, with a median age of 34 years. The 49 patients (57.0%) had stage I/II tumors, and 36 patients (41.9%) had stage III/IV tumors. The majority of tRCC cases showed International Society of Urological Pathology (ISUP) grade 2 (*n* = 36, 41.9%) and grade 3 (*n* = 40, 46.5%), and the rest cases showed grade 4 (*n* = 10, 11.6%) (Supplementary Data 1). All cases were preliminarily screened by histopathologists and diagnosed by fluorescence in situ hybridization (FISH)[31], and 68 cases (79.1%) had their gene fusion types confirmed using next-generation sequencing (NGS) (Supplementary Data 1, Methods). There were five cases of *TFEB*-tRCC, rarer than the *TFE3*-tRCC (*n* = 63) cases, which were consistent with previous reports[1]. We identified 4 fusion partners of *TFEB*, including *CLTC*, *ACTB*, *NEAT1*, and *EIF4A2*. Fifteen fusion partners of *TFE3* were identified in this cohort, among which *ASPSCR1*

(*n* = 21), *SFPQ* (*n* = 14), *NONO* (*n* = 6), *PRCC* (*n* = 6), *MED15* (*n* = 4), and *LUC7L3* (*n* = 3) were recuring *TFE3* fusion partners (Fig. 1b). *PTPN12*, *ZNF627*, *EWSR1*, *PARP14*, *KHSRP*, *MATR3*, *RBM10*, *U2AF2*, and *VCP* were the rare *TFE3* fusion partners observed in this cohort. The *PTPN12-TFE3*, *ZNF627-TFE3*, *U2AF2-TFE3* and *EIF4A2-TFEB* fusion types of tRCC were also observed (Supplementary Fig 1a, b). In addition, we found two cases of tRCC with two fusion types (Fig. 1b, Supplementary Data 1).

WES data of NATs were used as a reference to detect genetic variants in this cohort. The mean sequencing coverage in the hg38 reference genome was 118.91× for tumor tissues and 60.96× for NATs (Supplementary Fig. 1c). Among the 86 tumors and 84 paired NATs, we detected 2,563 non-silent mutations and 19,616 silent mutations. MutsigCV[32] was used to identify the significantly mutated genes (SMGs) in the tRCC. *BCDIN3D*, *NDRG1*, *ZNF668*, and *GNPTG* were identified as the SMGs in tRCC (Fig. 1c). Among the SMGs in tRCC, *NDRG1* acts as a tumor suppressor and plays an important role in p53-mediated caspase activation and apoptosis. The functions of *BCDIN3D*, *ZNF668*, and *GNPTG* in tumor are poorly studied. Commonly mutated genes in other RCCs, such as *VHL*, *PBRM1*, *BAP1*, and *MET* were not detected in our tRCC samples. In addition, we found the tumor suppressor genes *TP53* (*n* = 3), *BRCA1* (*n* = 3), *TSC2* (*n* = 3), *SETD2* (*n* = 3), *KMT2D* (*n* = 3), and oncogene *IRS2* (*n* = 3) recurrently mutated in tRCC (Supplementary Data 2).

For the proteomics and phosphoproteomics data analysis, Pearson's correlation coefficients were calculated for all quality control (QC) runs of the HEK293T cell samples. The median correlation coefficient among the QC samples was 0.91 for proteomics QC and 0.96 for phosphoproteomics QC, respectively (Supplementary Fig. 1d, e), which demonstrated the consistency and stability of the mass spectrometry platform. Proteomics analysis of all samples measured a total of 14,073 proteins (Fig. 1d), among which 11,471 proteins were common between tumor tissues and NATs, whereas 1487 and 1115 proteins were detected specifically in tumor tissues and NATs, respectively. On average, we identified 5607 proteins per sample, ranging from 2837 to 8120, in which more than 90% of samples had identified protein numbers over 4000. Proteome quantification was conducted using the iBAQ algorithm, followed by normalization to fraction of total (FOT) as reported previously (Supplementary Fig. 1f, g). Furthermore, a total of 33,853 phosphosites, corresponding to 6469 phosphoproteins, were identified (Fig. 1e).

RNA sequencing (RNA-seq) analysis identified 13,313 genes with fragments per kilobase of transcript per million fragments mapped (FPKM) of more than 1 (Supplementary Fig. 1h, i), allowing us to explore the relationship between the transcriptome and full proteome. The mRNA-protein correlation was moderate with sample-wise median Spearman's correlation of 0.39 (Supplementary Fig. 1j), consistent with the previous report[33]. For tumor tissues, 7206 mRNA-protein pairs were detected, and the median gene-wise Spearman's correlation between protein and mRNA was 0.059. Gene set enrichment analysis (GSEA) revealed that genes with positive mRNA-protein correlation were enriched in kidney elevated proteins (kidney signature), Gly/Ser/Thr metabolism, and extracellular matrix (ECM) receptor interaction, whereas genes with negative correlation were enriched in proteasome and oxidative phosphorylation (OXPHOS) (Fig. 1f). The poor correlation of mRNA-protein pairs in OXPHOS was also observed in a variety of other tumors, such as ccRCC[34].

The transcriptomic similarity was used to infer the origin of renal malignancies, including chromophobe RCC (ChRCC) and renal medullary carcinoma, in previous reports[35,36]. We computed the global inter-profile correlation of ccRCC, papillary RCC (PRCC), ChRCC, tRCC, and different microdissected nephron regions. The results showed that tRCC mRNA expression have a high degree of correlation with the proximal tubule (Supplementary Fig. 1k), indicating that tRCC, similar to ccRCC and PRCC, originated from proximal tubule.

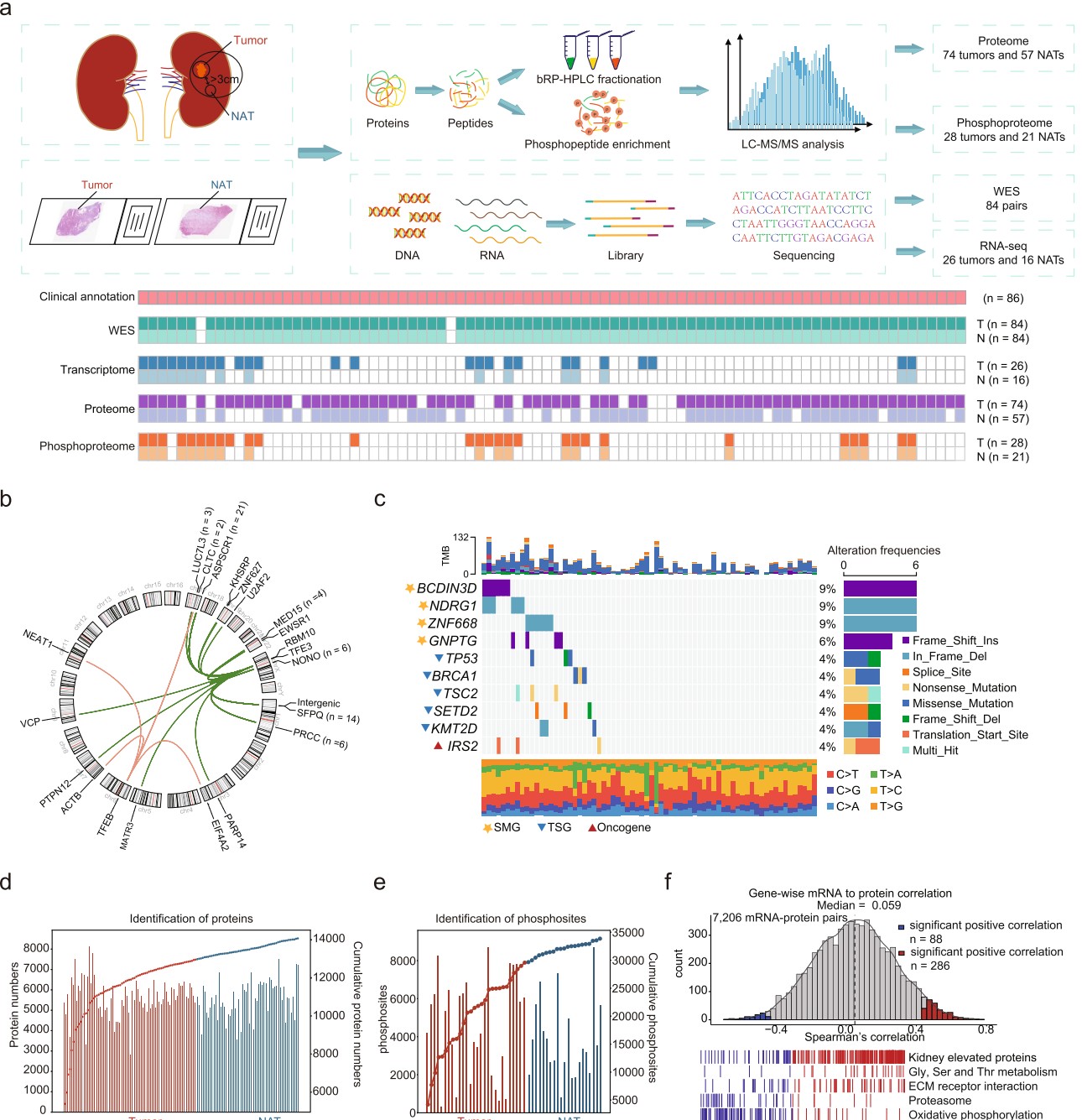

**Fig. 1 | Molecular profiling of MiT family tRCC. a** Schematic representation of tRCC multiomics analyses, including WES, RNA-seq, proteomics, and phospho-proteomics. **b** Circos plot showing the TFE3/TFEB gene fusion events. **c** Genomic profile of 70 tRCC tumors with somatic mutations. SMGs, TSGs, and oncogenes are noted by different shapes. **d** Overview of proteomic profiles of tRCC samples. **e** Overview of phosphoproteomic profiles of tRCC samples. **f** Gene-wise mRNA-protein correlation and functional enrichment.

In summary, our study established a comprehensive landscape of MiT family tRCC at the genomic, transcriptomic, proteomic, and phosphoproteomic levels.

## Molecular alterations in tRCC tumors compared to adjacent tissues

Principle component analysis (PCA) revealed clear distinctions between tumor and adjacent tissues at the transcriptome, proteome, and phosphoproteome levels (Fig. 2a, b, Supplementary Fig. 2a). Interestingly, we found that upregulated TFE3 was only observed at the protein level and not at the mRNA level in tRCC tumor tissues (Fig. 2c). To estimate *TFE3* activity in the tRCC samples, we collected

transcription factor targets of *TFE3* from DoRothEA[37]. Single-sample GSEA (ssGSEA)[38] revealed that inferred *TFE3* activity, based on *TFE3* target abundances using both mRNA and proteome data, was significantly upregulated in tumors (Supplementary Fig. 2b, c). *TFE3* activities were negatively correlated with the kidney signature (Supplementary Fig. 2d). Correspondingly, the kidney signature scores were downregulated in tumors, suggesting that *TFE3* activity plays an important role in the loss of kidney identity in tRCC carcinogenesis.

At the proteome level, we identified 1727 proteins that showed significant differential expression (fold-change [FC] > 2; Benjamini–Hochberg adjusted *p* < 0.05), with 836 proteins

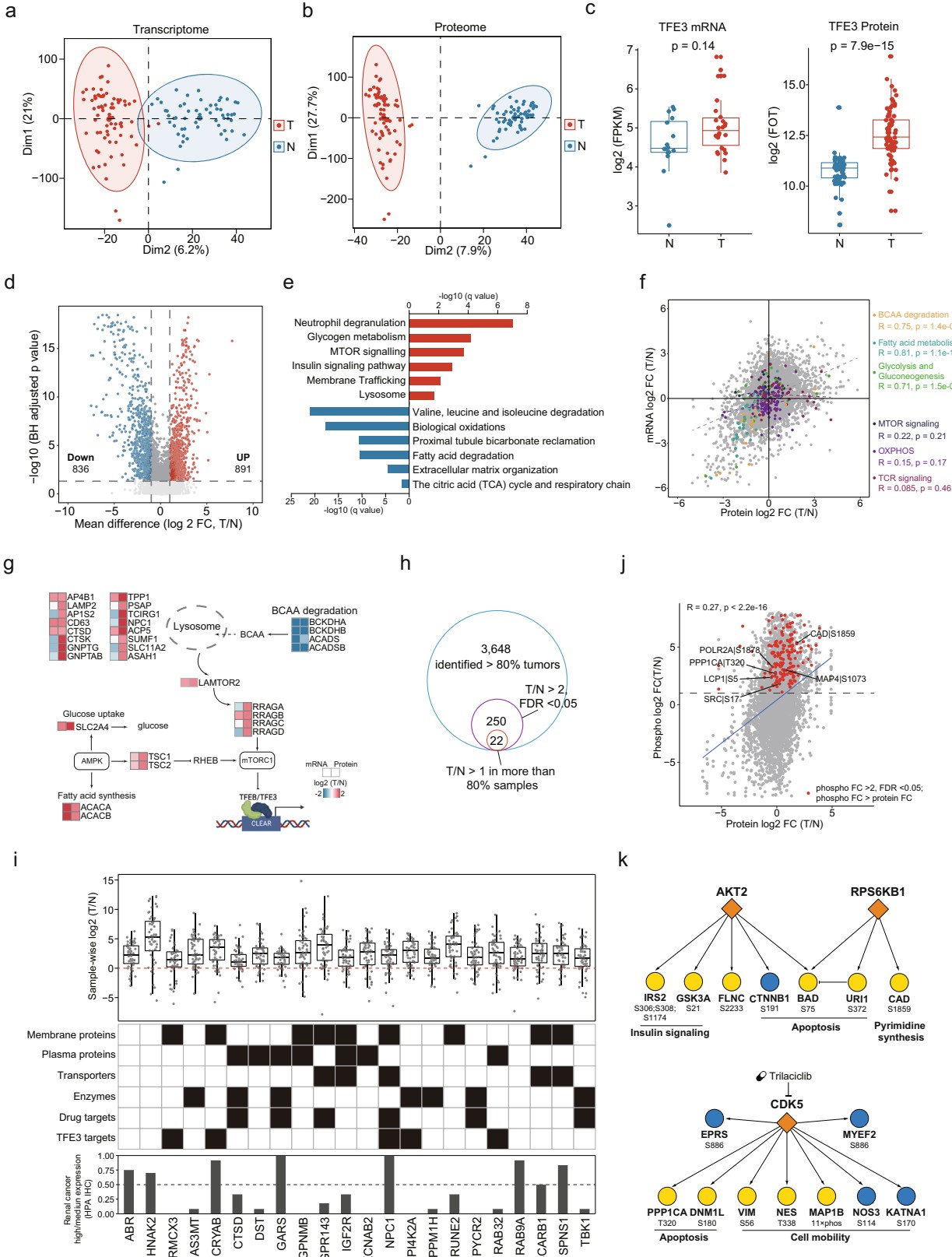

downregulated and 891 upregulated in tRCC tumors compared to adjacent tissues (Fig. 2d; Supplementary Data 4). Enrichment analysis revealed neutrophil degranulation, insulin signaling pathway, glycogen metabolism, membrane trafficking, mTOR signaling, and lysosomal function to be upregulated in tumors. Extracellular matrix organization, proximal tubule bicarbonate reclamation, and multiple metabolic pathways (valine, leucine, isoleucine degradation, fatty acid degradation, biological oxidation, the citric acid [TCA] cycle, and respiratory chain) were found to be downregulated ($q < 0.05$) (Fig. 2e).

At the transcriptome level, 1,626 tumor-upregulated genes were enriched in immune response, extracellular matrix organization, cell cycle, and apoptosis. 1,247 tumor-downregulated genes were enriched

**Fig. 2 | Molecular alterations in tRCC tumors compared to adjacent tissues.**
**a**–**b** Global transcriptome and proteome PCA plots. Red, tumor; Blue, NATs.
**c** Boxplots of TFE3 gene product levels displaying discordant mRNA (N, n = 16; T, n = 26)-protein (N, n = 57; T, n = 74) expression. Boxplots show the median (central line), the 25–75% interquartile range (IQR) (box limits), the ±1.5 × IQR (whiskers). P values are derived from two-sided Wilcoxon rank-sum test. **d** Volcano plot showing DEPs (Benjamini–Hochberg-adjusted p value < 0.05, FC > 2) in tumor and normal adjacent tissues. **e** DEPs in tumors and adjacent tissues, and their enriched biological pathways. **f** Scatterplots depicting expression of mRNA (x axis) and protein (y axis). Genes involved in BCAA degradation, Fatty acid metabolism, Glycolysis and Gluconeogenesis, MTOR signaling, OXPHOS, and TCR signaling were indicated by different colors. **g** Schema of uncoupling of MTOR pathways at mRNA and protein levels. **h** The pipeline for tRCC biomarker identification. **i** Log2-fold-change between tumor and matched NATs (n = 54) is shown for the 22 tRCC biomarkers. Boxplots show the median (central line), the 25–75% IQR (box limits), the ±1.5 × IQR (whiskers). These biomarkers are annotated with potential clinical utilities and IHC staining scores defined by HPA. **j** Comparison of abundance changes between phosphosites and their corresponding proteins. Red points indicate the phosphosites with >2-fold increase (Benjamini–Hochberg adjusted p < 0.05) and change stronger than in the corresponding protein. Phosphosites with functional annotations are indicated. **k** The kinase-substrate links of significantly activated kinases (at both protein abundance and kinase activity).

in multiple metabolic pathways (Supplementary Fig. 2e, f). Analysis of the differential abundances of mRNA and protein levels between tumors and adjacent tissues revealed consistent alterations in multiple metabolic pathways (including amino acid, fatty acid, and glucose metabolism) at mRNA and protein levels (Fig. 2f, Supplementary Fig. 2g). However, there was remarkable uncoupling of mRNA and protein expression in mTOR signaling and OXPHOS (Fig. 2f, g). Notably, uncoupling of mRNA and protein expression of OXPHOS was also observed in ccRCC[34]. Many studies have demonstrated the activation of mTOR signaling in tRCC[39,40], highlighting the importance of tRCC research using the proteomic approach. After the rigorous screening of proteomic data, we nominated 22 candidate tRCC biomarkers (Fig. 2h, i, Supplementary Data 4), which may aid in tRCC diagnosis. The potential clinical utility of these protein markers is annotated in Fig. 2i, and immunohistochemistry (IHC) staining data in The Human Protein Atlas (HPA) were used to help eliminate nonspecific RCC markers. Among the 22 candidate biomarkers, ten showed high or medium IHC scores in less than 20% of common kidney cancer samples in the HPA dataset, seven are plasma proteins, and three are secreted proteins. Glycoprotein nonmetastatic B (GPNMB) was identified as a diagnostic marker for TFE3-tRCC in a previous study[41], supporting the reliability of our data. In addition, two candidates (DST, TBK1) from these ten proteins, less expressed in kidney cancer samples, were validated by IHC staining. The results showed that tRCC tumors presented stronger DST and TBK1 IHC staining than ccRCC, PRCC, ChRCC, and normal kidney tissue (Supplementary Fig. 2h), indicating DST and TBK1 were potential biomarkers to distinguish tRCC and other kidney malignancies.

As for the phosphosites, 221 were significantly upregulated in tumor tissues compared with adjacent tissues (FC > 2, Benjamini–Hochberg adjusted p < 0.05). Among these, 182 changed stronger than in the corresponding protein (Fig. 2j), and only 6 sites had known functional annotations[42]. CAD S1859 phosphorylation promoted cell proliferation through its control of the de novo synthesis of pyrimidines[43]. Phosphorylation of MAP4 S1073 led to a pronounced dynamic instability in microtubules[44]. LCP1 S5 phosphorylation increased the invasive capacity of cells[45]. Kinase activities were also inferred based on the levels of substrate phosphorylation by kinase-substrate enrichment analysis (KSEA)[46]. Three kinases (AKT2, RPS6KB1, CDK5) showed significantly increased activity and protein abundances (Supplementary Fig. 2i, Supplementary Data 4). The kinase-substrate links are showed in Fig. 2k. Activation of RPS6KB1, an essential effector of mTOR signaling, further supported the mTOR pathway as a potential therapeutic target for tRCC. Trilaciclib, an FDA-approved drug[47] that targeted CDK5, is a potential treatment for tRCC.

### The Landscape of Mutational Signatures and Proteomic Impact
We compared the tumor mutational burden (TMB) and chromosome instability (CIN) scores of tRCC with other major RCC subtypes, including ccRCC, PRCC, and ChRCC (Supplementary Fig. 3a, b). It was shown that tRCC had lower TMB compared with ccRCC and PRCC, and it had higher TMB compared with ChRCC (Supplementary Fig. 3a). Interestingly, tRCC showed the lowest chromosome instability among

all the RCCs (Supplementary Fig. 3b). To determine the contribution of endogenous and exogenous mutagens to genetic alterations in tRCC, we decomposed the mutation spectra using non-negative matrix factorization, and four mutational signatures were identified (Fig. 3a). Cosine similarity analysis was performed to match the extracted signatures to the COSMIC reference signatures[48]. The mutational signatures that best matched those in the tRCC tumors were SBS26, SBS6, SBS40, and SBS22. Signatures SBS6 and SBS26 were correlated with defective DNA mismatch repair (MMR). SBS22 was associated with exposure to aristolochic acid, a type of carcinogen from traditional Chinese herbs[49,50]. All four mutational signatures were significantly correlated with TMB, while only Sig2 (SBS6) was significantly correlated with CIN (Fig. 3b). It was discovered that SBS6 and CIN were associated with poor progression free survival (PFS) (Fig. 3c). To investigate the cause of SBS6 at the protein level, we found that DNA repair associated proteins ATM, DDB1, PARP1, XRCC5 and CUL4A were significantly downregulated in tumors with SBS6 (Fig. 3d), instead of the commonly reported MMR proteins MLH1, MSH2, MSH3, and MSH6 (Supplementary Fig. 3c). Moreover, DDB1 abundance was negatively correlated with CIN (Supplementary Fig. 3d) and positively correlated with PFS (Supplementary Fig. 3e), indicating that reduced DDB1 might be an important cause of SBS6 associated CIN.

Next, we investigated the effects of the SBS6 signature at the proteome level. We found that SBS6 was negatively correlated with OXPHOS and positively correlated with oxidative damage (Fig. 3e). A major function of the mitochondria is the synthesis of adenosine triphosphate (ATP) by OXPHOS, and impaired OXPHOS is an important sign of mitochondrial dysfunction. Thus, we used OXPHOS as the indicator of mitochondrial function. It was reported that mitochondrial dysfunction results in elevated reactive oxygen species (ROS) levels, which induces cellular oxidative damage[51]. We consistently found that impaired OXPHOS was correlated with increased oxidative damage in tRCC tumors (Fig. 3f). Impaired glutathione (GSH) synthesis-related enzymes (GSR, GSS, GCLC) in SBS6 tumors further reinforced the association of SBS6 with oxidative damage in tRCC (Fig. 3g). In addition, patients with a higher expression of Glutathione-Disulfide Reductase (GSR) appeared to have better prognostic outcomes (Supplementary Fig. 3f). As SBS6 was also positively correlated with several immune pathways, such as the DNA damage bypass, cytosolic DNA sensing, and interferon-alpha response pathways (Fig. 3e), we deconvoluted the tumor microenvironment components using xCell[52]. SBS6 was positively correlated with the infiltration of CD8+ Tem, Tgd cells, CD4+ naïve T-cells, and CD4+ Tcm cells (Supplementary Fig. 3g). CD4+ Tcm cells were also negatively correlated with OXPHOS and PFS (Supplementary Fig. 3h, i). In conclusion, in tRCC tumors with SBS6 signatures, mitochondrial dysfunction-induced ROS, impaired GSH synthesis, defective DNA repair, and increased DNA damage caused a latent self-propagating cycle of further damage (Fig. 3h).

### Somatic Copy Number Alterations and their Proteomic Consequences
Somatic Copy Number Alterations (SCNAs) in tRCC were identified using GISTIC[53] (Supplementary Data 2). We identified 14 significant

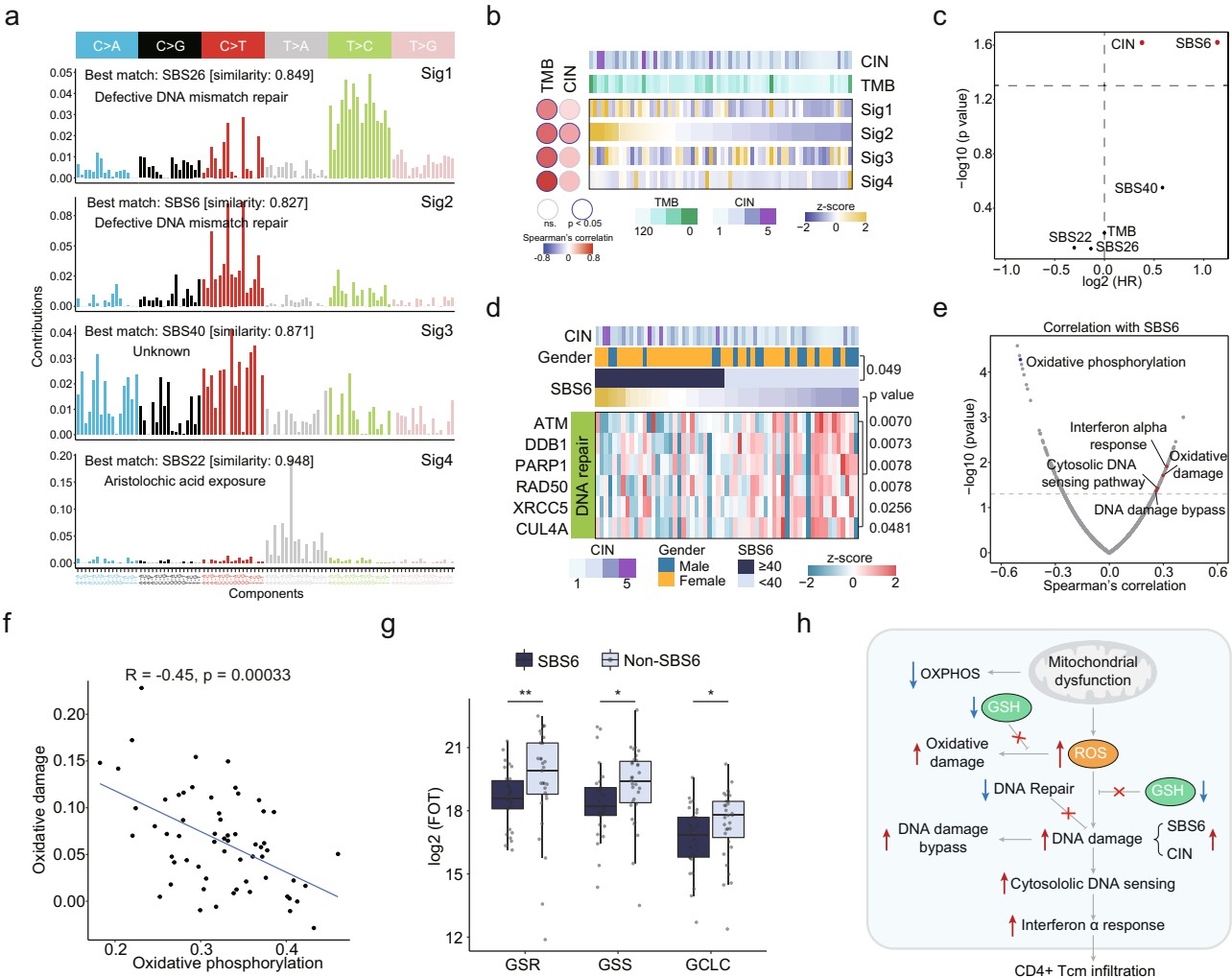

**Fig. 3 | Identification of mutational signatures and associated signaling pathways. a** Decomposition of four mutational signatures from the tRCC cohort. **b** Correlations between mutational signatures and TMB and CIN. **c** Cox regression analysis of TMB, CIN, and mutational signatures. **d** Heatmap showing differentially expressed DNA repair molecules between tRCC with and without SBS6 (two-sided Wilcoxon rank-sum test). **e** Scatter plot showing the correlation of SBS6 and Wikipathway ssGSEA scores. **f** Wikipathways oxidative damage and oxidative phosphorylation shows a strong negative correlation. **g** Relative protein abundance

in SBS6 (*n* = 30) and non-SBS6 (*n* = 31) groups for ROS defense proteins GSR (**\*\****p* = 0.0065), GSS (**\****p* = 0.019), and GCLC (**\****p* = 0.044). Boxplots show the median (central line), the 25–75% IQR (box limits), the ±1.5 × IQR (whiskers). *P* values are derived from two-sided Wilcoxon rank-sum test. GSR Glutathione-Disulfide Reductase, GSS Glutathione Synthetase, GCLC Glutamate-Cysteine Ligase Catalytic Subunit. **h** Schematic representation of the molecular features of tRCC with SBS6 signature.

---

arm-level SCNAs, including amplification of 1q, 5p, and 5q, and deletions of 1p, 3p, 4q, 6q, 9p, 9q, 18p, 18q, 19p, 19q, and 22q in tRCC (Fig. 4a). Focal SCNAs are shown in Fig. 4b. Cytobands in chromosome 1p (1p13.2, 1p36.12, 1p36.21) contained the most frequently deleted focal regions, and 1q21.1 was the only focal amplification event (Fig. 4b). Deletion events were more frequent than amplification events in both arm and focal levels (Supplementary Fig. 4a, b).

Next, we examined the correlations of SCNAs with mRNA and protein abundance (Supplementary Fig. 4c, d). The *trans*-acting SCNA hotspots were identified on 1p, 2q, 3p, 8p, 12q, 14q at the proteome level (Supplementary Fig. 4d). The mRNA expression levels showed significant *cis*-effect were enriched in fatty acid beta-oxidation. The proteins that showed significant *cis*-effect were enriched in valine, leucine, and isoleucine degradation, neutrophil degranulation, ketogenesis, and fatty acid beta-oxidation (Supplementary Fig. 4e). We used Cox regression to identify associations between significant CNA events and clinical outcomes (Fig. 4c). Deletions of 6q, 18p, 18q, 3p, 16p, and 1p, and amplification of 1q and 5p were associated with poorer PFS. Multivariate analysis was performed using these arm-level CNAs,

which showed that deletions of 6q and 3p and amplification of 1q were the dominant CNA events associated poor PFS (Supplementary Fig. 4f, g, Supplementary Data 2). Moreover, deletions of 6q and 3p were correlated with decreased overall survival (OS) (Supplementary Fig. 4f, g). Earlier genomic studies of tRCC indicated that 17q gain and 9p loss were significantly correlated with poor outcomes[29,30]. We compared the survival curves of patients with and without 17q gain and 9p loss and found that our results were consistent with previous reports (Supplementary Fig. 4h).

As chromosome 3p was a *trans*-acting SCNA hotspot and 3p deletion was correlated to poorer clinical outcomes (Supplementary Fig. 4f), the proteome impacts of 3p deletion were further surveyed. GSEA revealed that the proteins that had positive correlations with the 3p copy number (CN) were converged on aggrephagy, while the proteins that had negative correlations with the 3p CN were converged on complement and coagulation cascades (Fig. 4d, Supplementary Fig. 4i). As for the *cis*-effect, abundances of ATG7, an E1-like activating enzyme essential for autophagy, were significantly correlated with the 3p CN (Spearman's correlation, *p* = 7.87E-4). Moreover, proteins

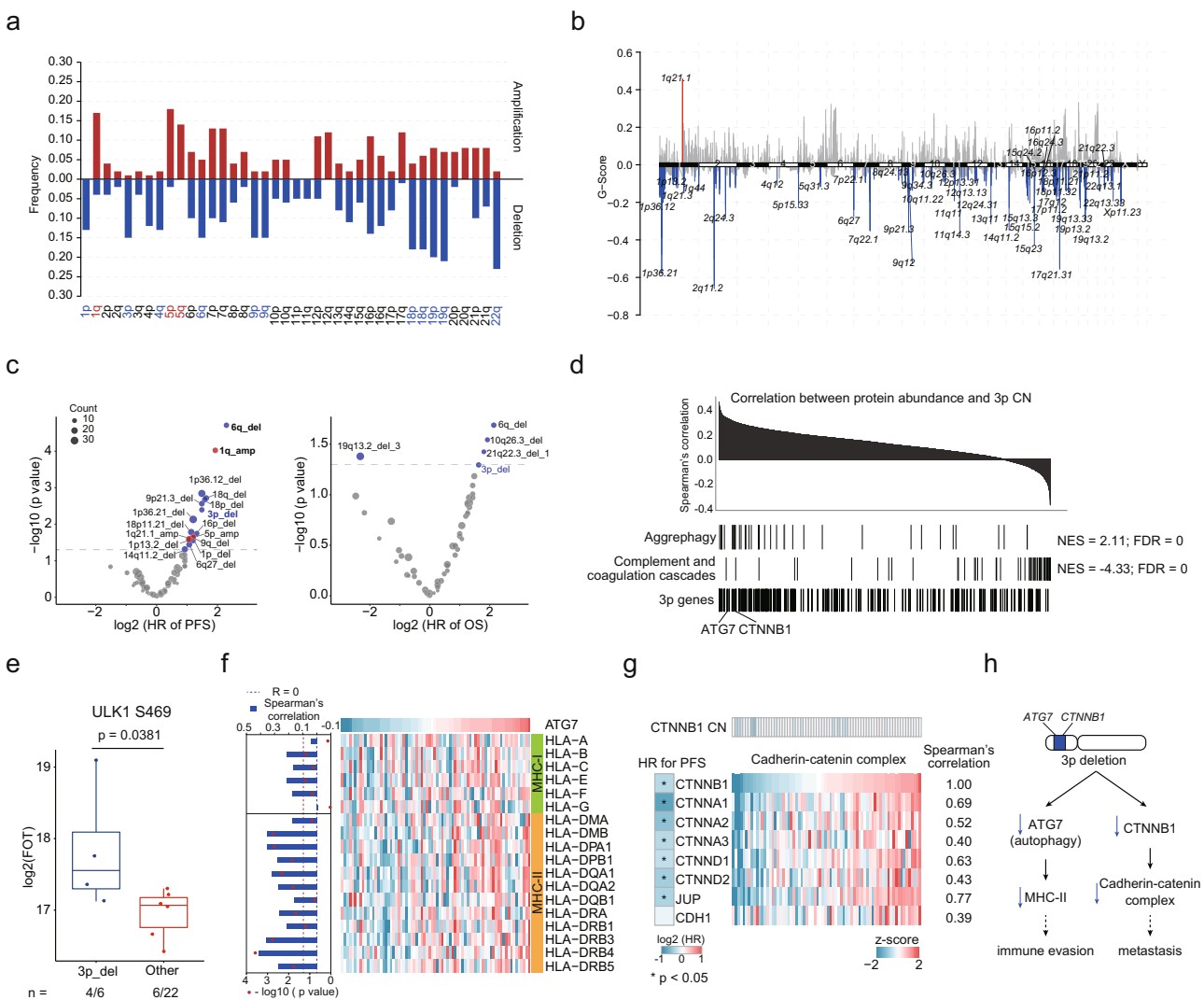

**Fig. 4 | Somatic copy number alteration analysis in tRCC cohort. a** Arm-level SCNA events. Red denotes amplification and blue denotes deletion. Significant events are highlighted using red and blue ($q < 0.10$). **b** Focal SCNA events. Focal peaks with significant copy number amplifications (red) and deletions (blue) ($q < 0.05$) are shown. **c** Cox regression analysis of significant arm-level CNA and focal CNA events. **d** Identifying *cis*- and *trans*-effect of 3p deletion. Proteins are ranked based on the correlation between protein abundance and 3p CN. **e** *Trans*-effect of 3p deletion on ULK1 S469 phosphorylation level. Boxplots show the median (central line), the 25–75% IQR (box limits), the ±1.5 × IQR (whiskers). *P* value is derived from two-sided Wilcoxon rank-sum test. **f** Correlations between ATG7 protein abundance and MHC molecules. **g** *Cis* and *trans*- effects of CTNNB1 deletion on cadherin-catenin complex abundances. Cox regression analysis of cadherin-catenin complex abundances are shown in left. **h** A model depicting the association of 3p deletion and poor prognosis in tRCC.

involved in autophagy, such as ATG4B, PIK3C3, HSPA8, PARK7, and ubiquitin, were also downregulated in tRCC tumors with 3p deletion (Supplementary Fig. 4i). ULK1 S469 phosphorylation, reported to reduce the occurrence of autophagy[54], was upregulated in 3p deletion tumors ($p = 0.0381$) (Fig. 4e), indicating that the deletion of 3p in tRCC tumors might result in the impairment of autophagy. As autophagy was reported to play an important role in antigen presentation[55,56], we investigated the correlation between ATG7 and antigen presentation mechanisms. We observed that ATG7 abundance was positively correlated to abundances of MHC-II molecules, but not MHC-I molecules (Fig. 4f). Another important *cis*-effect in 3p occurred in *CTNNB1*, a part of the cadherin-catenin complex. The abundance of CTNNB1 was significantly correlated with other components of the cadherin-catenin complex (Spearman's correlation, $p < 0.05$). Downregulation of most components of the cadherin-catenin complex was associated with poorer PFS (Fig. 4g). It was reported that the cadherin-catenin complex performs a key role in cell adhesion. Loss of cell adhesion is seen as a key step in the development of tumor metastasis. Altogether, 3p

deletion was associated with immune evasion and metastasis owing to the *cis*-acting elements on *ATG7* and *CTNNB1* respectively (Fig. 4h).

## The proteomic differences among the tRCC fusion types
Despite both being classified as MiT family tRCC, *TFE3* fusion tumors were relatively more aggressive, while *TFEB* fusion tumors had a better prognosis[1,57]. By comparing these two TFE fusion types, 20 proteins were identified to be upregulated in *TFE3* fusion tumors, and 518 proteins were upregulated in *TFEB*-tRCC (Fig. 5a, Supplementary Data 5). *TFEB* was particularly overexpressed in *TFEB*-tRCC (FC = 27.34, $p = 0.0029$) (Fig. 5a, b). Accordingly, TFEB-inferred activities were also stronger in *TFEB*-tRCC than in *TFE3*-tRCC (Fig. 5b). However, TFE3 abundances and inferred activities did not show significant differences (Fig. 5c). Enrichment analysis revealed the upregulated glycolysis, gluconeogenesis, neutrophil degranulation, endocytosis, necroptosis, and membrane trafficking in *TFEB*-tRCC. Ca-calmodulin-dependent protein kinase activation was upregulated in *TFE3*-tRCC (Fig. 5d).

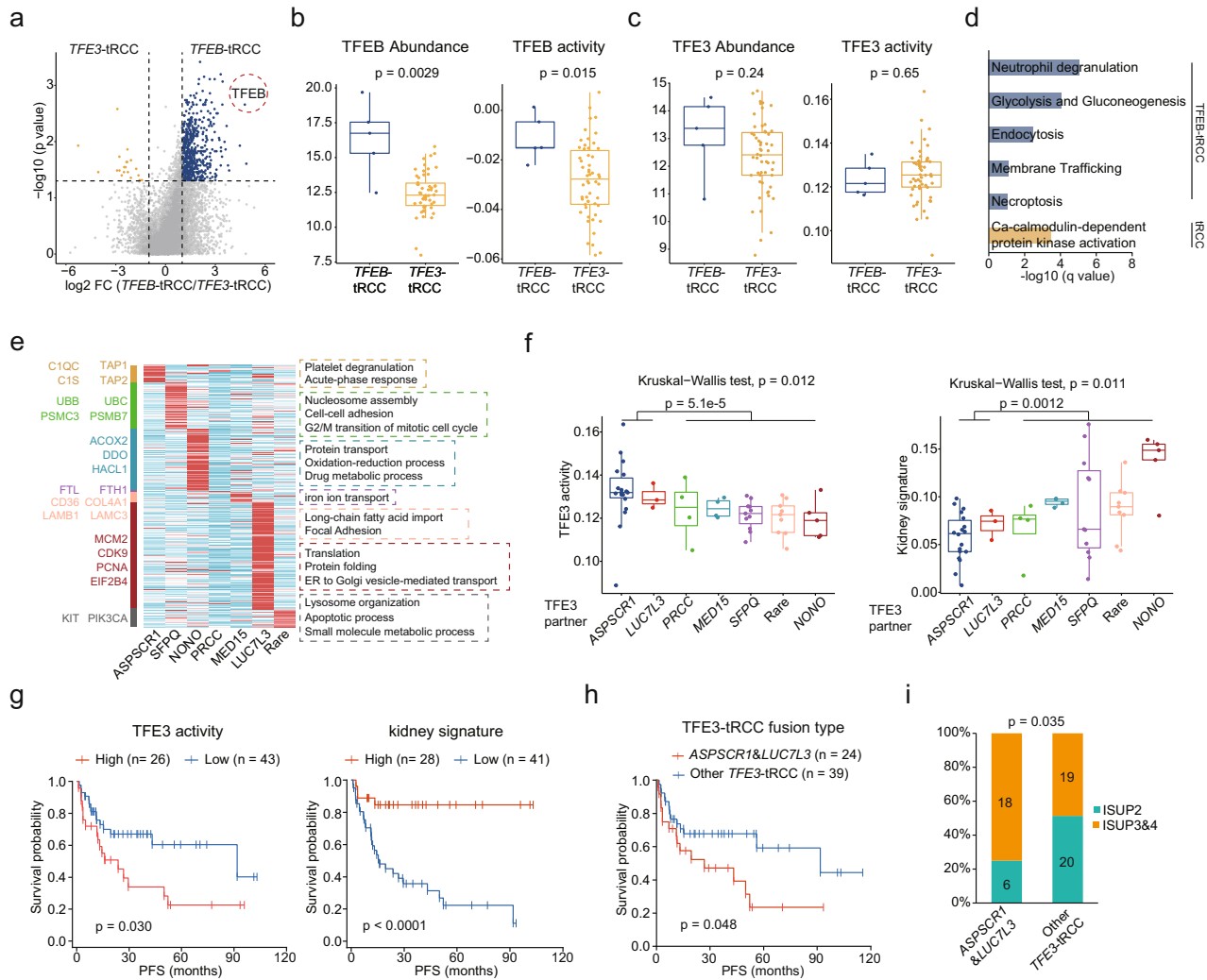

**Fig. 5 | Molecular heterogeneity of different fusion types of tRCC. a** Volcano plot showing DEPs (two-sided Wilcoxon rank-sum test, *p* value <0.05, FC > 2) of *TFEB*-tRCC and *TFE3*-tRCC tumors. **b, c** Comparisons of TFEB and TFE3 product levels and activities in *TFEB*-tRCC (*n* = 5) and *TFE3*-tRCC (*n* = 54) tumors. **d** DEPs in *TFEB*-tRCC and *TFE3*-tRCC tumors enriched biological pathways. **e** Elevated proteins in different *TFE3*-tRCC fusion types and involved biological processes. **f** Comparison of TFE3 activities and kidney signature scores among different *TFE3*-tRCC fusion types (*APSCR1*, *n* = 18; *LUC7L3*, *n* = 3; *PRCC*, *n* = 4, *MED15*, *n* = 4; *SFPQ*, *n* = 11; Rare, *n* = 9; *NONO*, *n* = 5). *P* values are derived from two-sides Wilcoxon rank-sum test.

**g** Kaplan–Meier curves of PFS for patients with different TFE3 activities and kidney signature scores (two-sided log-rank test). **h** Kaplan–Meier curves of PFS for *ASPSCR1* and *LUC7L3 TFE3*-tRCC tumors versus other *TFE3*-tRCC tumors in this cohort (two-sided log-rank test). **i** Proportions of ISUP grades for *ASPSCR1* and *LUC7L3 TFE3*-tRCC tumors versus other *TFE3*-tRCC tumors (One-sided Fisher's exact test). Data in **b, c, f** are shown using boxplots. Boxplots show the median (central line), the 25–75% IQR (box limits), the ±1.5 × IQR (whiskers). *P* values are derived from two-sided Wilcoxon rank-sum test.

Different *TFE3* fusion partners were successively discovered in the previous decades[1], yet the molecular features of the different *TFE3* fusion types remained largely uncharacterized. We compared the abundances of *TFE3* and *TFE3* fusion partners among the different *TFE3* fusion types. The results showed that there was no significant difference in *TFE3* abundances among the different fusion types (Supplementary Fig. 5a). We next compared the abundances of different corresponding fusion partners, in which only *LUC7L3-TFE3* fusion upregulated the abundance of LUC7L3 (Supplementary Fig. 5a, b). To help discern biological insights stemming from the diverse *TFE3* fusion types of tRCC, we identified proteomic characteristics associated with *TFE3* fusions (Fig. 5e). In *ASPSCR1-TFE3* tRCC, we observed an upregulation of acute-phase response (C1S, C1QC) and antigen presentation (TAP1, TAP2). *SFPQ-TFE3* tRCC displayed elevations in the ubiquitin-proteosome system (UBB, UBC, PSMB7). *NONO-TFE3* tRCC had upregulated oxidation-reduction and drug metabolic processes (DDO, ACOX2). *PRCC-TFE3* had the least specifically upregulated proteins, which were enriched in iron ion transport. *LUC7L3-TFE3* tRCC

overexpressed proteins involved in the cell cycle (CDK9, PCNA, MCM2). *TFE3-tRCC* with a rare fusion partner and showed elevated apoptotic processes (PIK3CA).

Notably, we found that TFE3 activities and kidney signature scores differed among different tRCC fusions types (Fig. 5f, g). *ASPSCR1-TFE3* and *LUC7L3-TFE3* fusion tumors had higher TFE3 activities, while featured lower kidney signature scores, than other fusion types of *TFE3*-tRCC (Fig. 5f). As TFE3 activities and kidney signature scores were significantly associated with patient prognosis (Fig. 5g), *ASPSCR1* and *LUC7L3* fusion tumors were more aggressive than other fusion type of tumors (log-rank test, *p* = 0.048) (Fig. 5h). We further incorporated data on tRCC cases (*n* = 15) from the TCGA studies[58] with our data, getting a more significant difference (log-rank test, *p* = 0.033) (Supplementary Fig. 5c). Consistently, we found that *ASPSCR1* and *LUC7L3* fusion tRCC tumors have higher ISUP grade than other fusion types of *TFE3*-tRCC (Fig. 5i). To investigate how TFE3 affected clinical outcomes, we screened 12 TFE3 target proteins (Supplementary Fig. 5d). C1S, C3, and SERPING1, involved in complement cascades, were

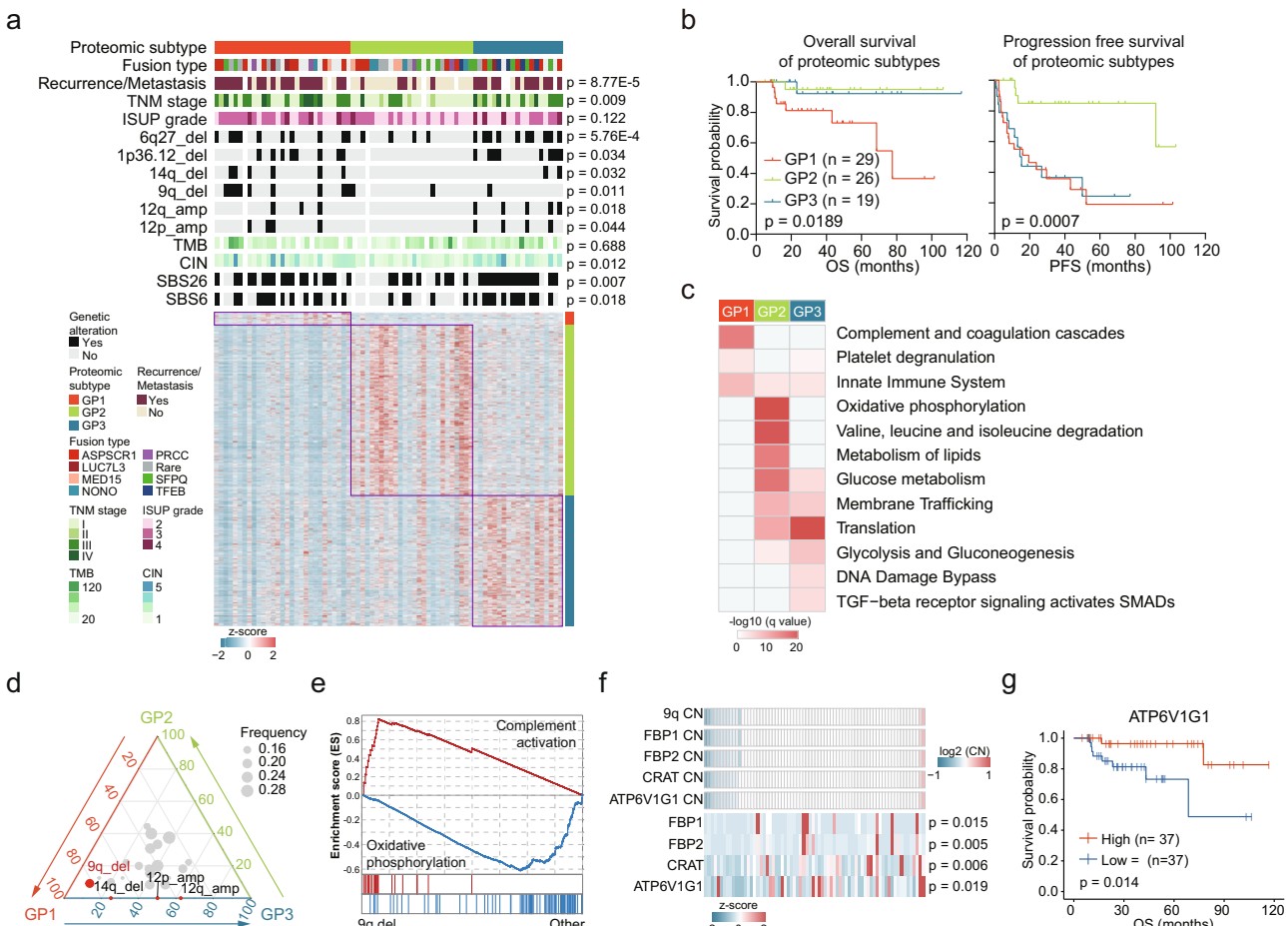

**Fig. 6 | Proteomic subtypes of tRCC and associations with genetic features and clinical outcomes. a** Relative abundances of upregulated proteins in the three proteomic subtypes and associations of proteomic subtypes with clinical and genetic features (Fisher's exact test or Kruskal–Wallis). **b** Kaplan–Meier curves of OS and PFS for the three subtypes (two-sided log-rank test). **c** Upregulated pathways enriched in the three proteomic subtypes. **d** Ternary plot showing the distribution of significant arm-level events in the three proteomic subtypes. CNA events with significant differences among three proteomic subtypes are indicated (Fisher's exact test). **e** GSEA plots showing the *trans*-effect of 9q deletion. **f** *Cis*-effects of 9q deletion in tRCC. *P* values are derived from two-sided Spearman's correlation test. **g** Kaplan–Meier curves of OS for patients with different ATP6V1G1 abundances (two-sided log-rank test).

---

upregulated by TFE3. ACAT1 and BDH1, which function in ketone catabolism, were downregulated by TFE3, indicating a ketogenic diet therapy in malignant tRCC[59]. Collectively, *TFE3* fused with diverse partners, and these fusions displayed differential TFE3 activities, further influencing protein expression patterns and clinical outcomes.

### Proteomic subtypes of tRCC

Due to the high degree of inter-tumoral heterogeneity, it was important to perform molecular subtyping of tRCC. We employed consensus clustering[60] to identify tRCC proteomic subtypes. The tRCC cases were classified into three subtypes, GP1, GP2, and GP3 comprising 29, 26, and 19 cases, respectively (Fig. 6a, Supplementary Fig. 6a–c). *TFEB*-tRCC showed overrepresentation in the GP3 tumors (Fig. 6a, Supplementary Fig. 6d). GP1 had the highest proportion of stage III&IV tumors, while GP2 had the highest proportion of stage I&II tumors (Fig. 6a). GP1 cases contained more high-grade (grade 3&4) tumors, compared to GP2&3 cases (Supplementary Fig. 6e). Remarkably, the proteomic subtypes significantly differed in OS (log-rank test, *p* = 0.0189) (Fig. 6b) and PFS (log-rank test, *p* = 0.0007) (Fig. 6b). Among the three subtypes, GP1 showed the shortest OS and PFS, and GP3 showed a fairly short PFS but a long OS (Fig. 6b). Correspondingly, 86.5% of the patients, who finally developed recurrence or metastasis, belonged to GP1 and GP3 (Fig. 6a, Supplementary Data 6).

We conducted an overrepresentation analysis of elevated proteins in each subtype (Fig. 6c). In total, 77, 1,009, and 630 proteins were upregulated in GP1, GP2, and GP3, respectively. Proteins upregulated in GP1 were enriched in pathways such as the complement and coagulation cascades (C4A, C7, C8A, C9), platelet degranulation (FGA, FGB, FGG), and the innate immune system (Fig. 6c, Supplementary Fig. 6f). GP2 was more associated with elevated metabolic pathways, including OXPHOS (COX4I1, UQCRB, SDHC), amino acid metabolism, lipid metabolism, and glucose metabolism (Fig. 6c, Supplementary Fig. 6f). Accordingly, GP2 had the highest kidney signature scores among the three subtypes (Supplementary Fig. 6g). GP3 had upregulated proteins that were enriched in pathways related to tumor proliferation and protein homeostasis, such as ribosome function, translation, and proteasome degradation (Fig. 6c, Supplementary Fig. 6f).

Next, we compared the genomic information of the proteomic subtypes established in this study to explore the driving effects of genetic alterations in the proteomic subtypes. TMB showed no significant differences among the three subtypes (Supplementary Fig. 6h). GP2 had the lowest CIN (Supplementary Fig. 6i). Mutational signatures also varied among the three subtypes (Supplementary Fig. 6j, k). GP2 had the lowest level of SBS6 and SBS26 (Supplementary Fig. 6k). As for arm-level CNA events, GP1 and GP2 showed the lowest similarity (Supplementary Fig. 6l), indicating arm-level CNAs

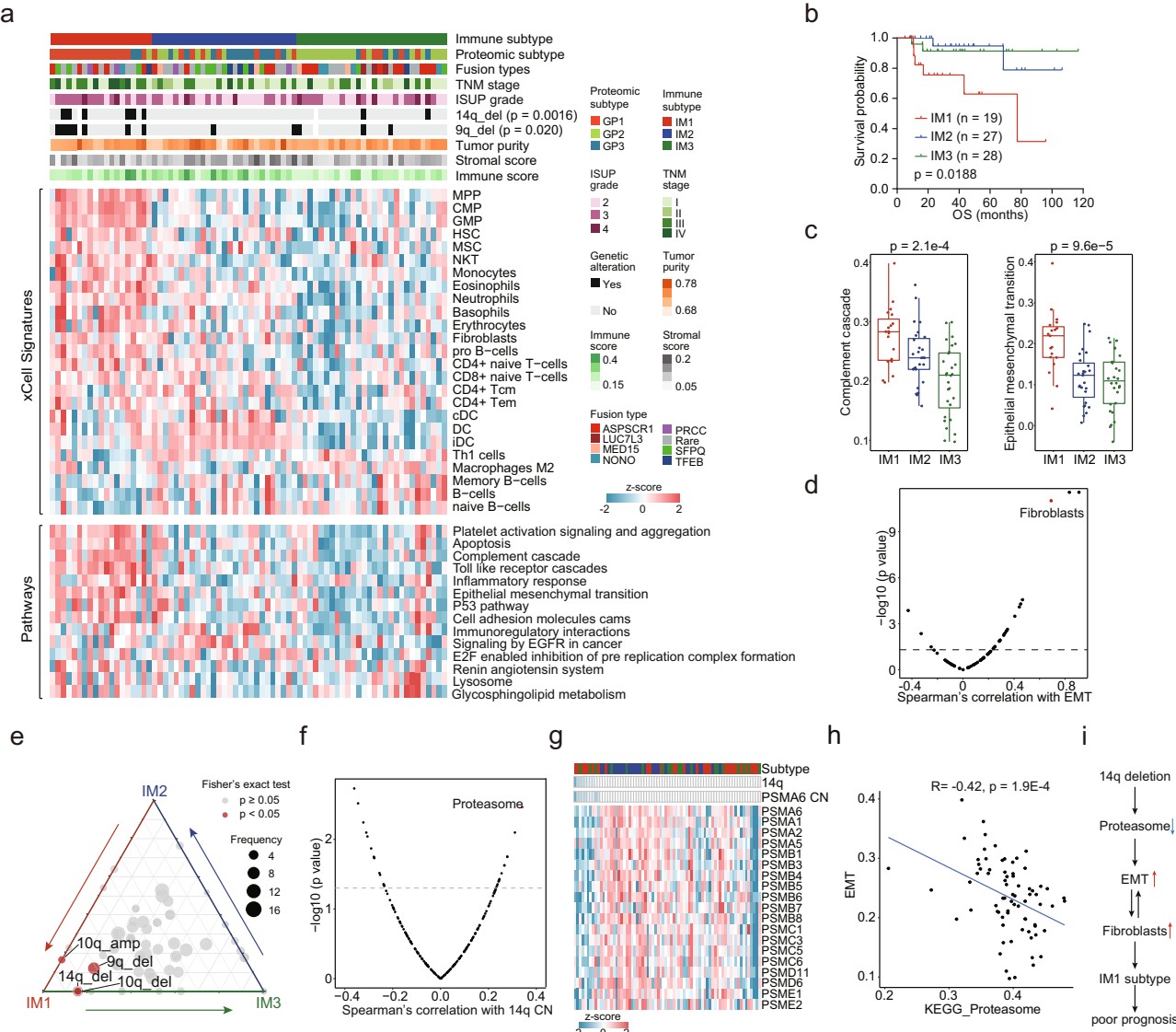

**Fig. 7 | Immune infiltration of tRCC tumors. a** Relative abundances of upregulated proteins in the three immune subtypes and associations of immune subtypes with clinical and genetic features and proteomic subtypes (Fisher's exact test). **b** Kaplan–Meier curves of OS for the three immune subtypes (two-sided log-rank test). **c** Comparison of pathway scores for the Complement Cascade and EMT among immune subtypes (IM1, $n = 19$; IM2, $n = 27$; IM3, $n = 28$). Boxplots show the median (central line), the 25–75% IQR (box limits), and the $\pm 1.5 \times$ IQR (whiskers). $P$ values are derived from Kruskal–Wallis test. **d** Scatter plot showing the correlation

of EMT scores and xCell immune signatures. Fibroblasts are indicated. **e** Ternary plot showing the distribution of significant arm-level events in the three immune subtypes. CNA events with significant differences among three subtypes are indicated (Fisher's exact test). **f** Proteasome showing the highest correlation with 14q CN. **g** *Cis* and *trans*-effects of 14q deletion on proteasome. **h** Correlation of proteasome scores and EMT scores (two-sided Spearman's correlation test). **i** A model depicting the association of 14q deletion and immune signature in tRCC.

profoundly impacted the proteome and clinical outcomes of tRCC. Deletions of 9q and 14q occurred more frequently in GP1, and amplifications of 12p and 12q were aggregated in GP1 and GP3 (Fisher's exact test, $p < 0.05$) (Fig. 6a, d). These four arm-level CNA events were significantly correlated with shorter PFS (Supplementary Fig. 6m). As deletion of 9q was the most significant CNA feature of GP1, we analyzed the proteomic impacts of 9q deletion in tRCC. GSEA showed that complement activation was enriched in tumors with 9q deletion, while OXPHOS was enriched in tumors without 9q deletion (Fig. 6e). *ATP6V1G1* (9q32), encoding a component of ATPase, showed significant CNA *cis*-effect (Fig. 6f). Additionally, *ATP6V1G1* abundances were significantly correlated with clinical outcomes (Fig. 6g). We hypothesized that deletion of 9q resulted in the downregulation of OXPHOS in both *cis*- and *trans*-acting manners, further impacted the patient clinical outcomes.

## Characterization of immune infiltration in tRCC

To investigate the characteristics of immune infiltration in tRCC, we performed cell-type deconvolution analysis using xCell[52]. Consensus clustering, based on inferred relative abundances of different cell types in the tumor microenvironment (TME), identified three immune subtypes (Fig. 7a, Supplementary Fig. 7a, b). The immune subtypes showed significant association with proteomic clusters (Fisher's exact test, $p = 4.43e-7$), reflecting the impact of TME composition on protein expression patterns (Fig. 7a). Consistent with proteomic clusters, the immune subtypes were significantly associated with clinical outcomes (Fig. 7b, Supplementary Fig. 7c), among which immune subtype 1 (IM1) had the lowest PFS and OS.

Stromal infiltration was similar among the three immune subtypes, while IM3 exhibited the lowest immune infiltration (Supplementary Fig. 7d). The IM1 was characterized by a high proportion of

metastatic tumors, elevated levels of granulocytes, progenitor cells, Natural killer T (NKT) cells, mesenchymal stem cells (MSC), fibroblasts, and monocytes. IM2 had the highest dendritic cells, and IM3 had the highest Th1 cells (Fig. 7a).

We found that complement cascade and Epithelial-to-mesenchymal transition (EMT) levels were enhanced in IM1 tumors (Fig. 7c, Supplementary Fig. 7e). It was reported that fibroblasts in the TME could enhance the EMT of tumor cells[61], which was also observed in our data (Fig. 7d, Supplementary Fig. 7f). We conducted immuno-fluorescence (IF) analysis to elaborate the association between fibroblasts and EMT state of tumor cell. The results showed that IM1 tRCC exhibited stronger co-staining of pan-cytokeratin (CK, the epithelial marker) and alpha-smooth muscle actin (α-SMA, the fibroblast marker), and stronger single staining of α-SMA than other subtypes of tRCC (Supplementary Fig. 7g), reflecting the enhanced EMT and fibroblast infiltration in IM1 tumors. In addition, It showed that IF co-staining of a large subset of tumors for CK and α-SMA (Supplementary Fig. 7g), demonstrating the presence of tumor cells undergoing EMT and a small subset of fibroblast in IM1 tRCC tumors microenvironment.

Notably, IM1 tumors had higher frequencies of 14q and 9q deletions (Fisher's exact test, $p < 0.05$), which were both correlated with poor prognosis (Fig. 7e, Supplementary Fig. 6m). As 75% of 14q deletion events were concentrated in IM1, we further analyzed the proteomic impact of 14q deletion. The proteosome was the most correlated pathway of 14q CN (Fig. 7f), which manifested through the significant correlations between 19 proteasome components and 14q CN (Fig. 7g). The abundance of PSMA6 (14q13.2), a cis-acting protein in 14q, was correlated with better prognosis (Supplementary Fig. 7h). As the proteasome played an important role in protein turnover, we surveyed the association between the proteasome and protein abundances in tRCC (Supplementary Fig. 7i). The results showed that the abundance of EMT-associated proteins and EMT scores were negatively correlated with the proteasome (Supplementary Figs. 7i, 7h), suggesting that the EMT was regulated by the proteasome in a post-translational mechanism in tRCC. In conclusion, 14q deletion regulated the EMT and IM1-like microenvironment in tRCC, which was associated with poor prognosis (Fig. 7i).

## Discussion

In this study, we present a comprehensive molecular analysis of tRCC, including genomics, transcriptomics, proteomics, and phosphoproteomics, investigating the molecular and clinical characteristics of this disease. Our results revealed fusion partners of TFE3 and TFEB, found correlations between fusion types and prognosis, identified the key genomic alterations in the development and progression of the disease, illustrated the functions of disease-related proteins, highlighted the biologic features of post-translational modifications, discovered molecular subtypes and immune subtypes, and suggested several potential therapeutic targets for the treatment of tRCC.

To provide molecular insights into tRCC carcinogenesis, we compared the expression patterns in tumors and NATs at the mRNA and protein levels. The results revealed that metabolic process dysfunction and the loss of kidney signature were the common features at both mRNA and protein levels, which is consistent with previous reports of ccRCC[34]. ccRCC was characterized as a genomic alteration-attributed metabolic disease, which was the same for tRCC tumors. More importantly, we found that mTOR signaling was upregulated in tRCC tumors at the proteome level. Phosphorylation signals in the mTOR signaling pathway were consistently activated in tRCC tumors. These results further supported mTOR signaling as a potential therapeutic target in treating tRCC. In a previous report, all patients that switched over to an mTOR inhibitor after vascular endothelial growth factor receptor-targeted therapy failure achieved stable disease[62]. However, as mTOR signaling was not abnormally activated in all patients, mTOR inhibitors displayed inconsistent patient

responses[62-64]. It was necessary to evaluate the efficiency of mTOR inhibitors in a larger cohort and determine whether mTOR signaling was activated in tumors before therapy.

All cancers originate from a single-cell that starts to behave abnormally due to acquired somatic mutations in its genome. Each mutational process may relate to the components of DNA replication, DNA damage repair, and DNA modification, and they generate a characteristic mutational signature[65]. Our analysis confirmed that mutation signature SBS6, correlated with defective DNA mismatch repair, was associated with the rapid progression of the disease. Integrated with the proteome data, we found that tRCC tumors with SBS6 signatures displayed mitochondrial dysfunction, impaired GSH synthesis, defective DNA repair, and increased DNA damage. Anti-tumor agents targeting synthetic lethality, like PARP inhibitors, which have been used in the treatment of DNA repair defective tumors, also have potential in tRCC therapy.

Previous studies have confirmed the correlation between common kidney cancer subtypes and cytogenetic alterations, such as the loss of 3p in ccRCC and trisomy 7 and/or 17 in PRCC[66,67]. However, the role of CNA events in tRCC remains unclear. Previous studies of tRCC indicated that 17q gain and 9p loss were the most common CNA events[29,30]. However, inconsistent with previous studies, our analysis identified 14 significant arm-level SCNAs, including amplification of 1q, 5p, and 5q, and deletions of 1p, 3p, 4q, 6q, 9p, 9q, 18p, 18q, 19p, 19q, and 22q. We hypothesize that such inconsistent results may be due to our relatively small sample size, heterogeneity of tRCC, and differences in the study populations. More importantly, in contrast with the previous studies, our study further explored the association of CNA and protein expression patterns and revealed that 3p deletion was correlated with the latent immune evasion and tumor metastasis by cis- and trans-effects.

It was interesting that TFE3 activity was similar amongst TFEB-tRCC and TFE3-tRCC tumors. We proposed two possible reasons. The first was that TFE3 protein levels were similar between TFEB-tRCC and TFE3-tRCC tumors. The second reason was TFEB fusion could mimic the enhanced TFE3 activity caused by TFE3 fusion on account of the highly conserved bHLH-LZ domains of MiT transcription factors[1]. It was observed that ASPSCR1 and LUC7L3 fusion TFE3-tRCCs were more aggressive than other fusion types of TFE3-tRCC. We ascribed this phenomenon to the higher TFE3 activities and lower kidney signature scores of ASPSCR1 and LUC7L3 fusion of TFE3-tRCC, comparing with other fusion types of TFE3-tRCC. Interestingly, ASPSCR1 and LUC7L3 were both located at 17q. Simultaneously, ASPSCR1 and LUC7L3 TFE3-tRCC contained more 17q amplification events, which was associated with poorer PFS. The impacts of fusion types on CNA events need to be further studied.

We performed molecular subtyping based on proteome data and immune subtyping based on xCell-deconvoluted TME components. Proteomic subtype GP1 and immune subtype IM1 showed great overlap in samples and similar aggressiveness and poor prognosis. It was noted that protein expression patterns impacted immune infiltration and microenvironment components impacted observed protein expression patterns[34]. Moreover, Braun et al. revealed the impact of CNA events on immune infiltration in ccRCC[68]. GP1 and IM1 showed significant enrichment of 9q and 14q deletion, revealing the interplay of CNAs, proteome pattern, and immune infiltration. Previous studies suggested TSC1 as a target of 9q deletion[68] and HIF1A as a target of 14q deletion[69]. In contrast, this study revealed that ATP6V1G1 at 9q and PSMA6 at 14q might be potential tRCC tumor suppressors. Although there was no significant association between TSC1 CN and protein abundances, it was notable that TSC1, a negative regulator of mTOR signaling located on 9q34.3, was focally lost in tRCC. Loss of TSC1 might contribute to mTOR signaling activation in tRCC.

In recent years, the use of immune checkpoint inhibitors has significantly improved the prognosis of advanced ccRCC[70,71]. Single-

cell RNA sequencing studies provided profound insight into the association of the TME and immune checkpoint blockade response in ccRCC[72-74]. Motzer et al. reported that atezolizumab combined with bevacizumab significantly improved the PFS of TFE fusion RCCs versus sunitinib[75]. In this study, the most aggressive immune subtype IM1 was characterized by enrichment of progenitor cells, granulocytes, fibroblasts, and monocytes in the context of inflammation. It was consistently reported that tumor-associated hematopoietic stem and progenitor cells (HSPCs) were associated with malignant and immunosuppressive phenotypes in glioblastoma[76]. Despite this study linking TME with tumor progression in tRCC, further studies are needed to establish the connection between immune cell compositions and clinical features.

This study has several limitations. Although our study had the largest number of patients to date investigating the multi-omic features of tRCC, the number was still insufficient, especially the *TFEB*-tRCC and *TFEB*-amplified RCC cases. More importantly, the underlying mechanism of tumorigenesis and development of tRCC remain unknown and warrant further investigation. The retrospective and single-center design of this study also led to several inherent biases, such as selection bias. In conclusion, despite these limitations, the proteogenomic analysis of tRCC, provided valuable insights into the biological underpinnings, disease diagnosis, prognosis assessment, and treatment selection of tRCC.

## Methods

### Clinical sample collection
The study was compliant with the ethical standards of Helsinki Declaration II and was approved by the institutional review board of Fudan University Shanghai Cancer Center (FUSCC) (050432-4-2108*). Written informed consent was obtained from each patient before any study-specific investigation was conducted.

In total, tumor and NAT samples were obtained from 86 eligible tRCC patients who had undergone nephrectomy at the Department of Urology of FUSCC. Median follow-up was 34.5 months (range, 5.1–116.9 months). This cohort was comprised by 33.7% ($n = 29$) males and 66.3% ($n = 57$) females, with a median age of 34 years. The 49 patients (57.0%) had stage I/II tumors, and 36 patients (41.9%) had stage III/IV tumors. The majority of tRCC cases showed International Society of Urological Pathology (ISUP) grade 2 ($n = 36$, 41.9%) and grade 3 ($n = 40$, 46.5%), and the rest cases showed grade 4 ($n = 10$, 11.6%). Other information was summarized in Supplementary Data 1. Histology sections were reviewed by an experienced genitourinary pathologist and all MiT family tRCC samples and confirmed by TFE3 break-apart fluorescence in situ hybridization (FISH) assay or next-generation sequencing (NGS).

### TFE3 and TFEB fusion determination
The DNA and RNA isolated tumor and NAT samples were used for TFE3 and TFEB fusion type determination by using the NGS based YuanSu450 gene panel (Shanghai OrigiMed Co., Ltd., Shanghai, China), which covers all the coding exons of 450 tumor-related genes that are frequently rearranged in solid tumors. The genes were captured and sequenced with a mean depth of 800× by using Illumina NextSeq 500.

Gene fusion/rearrangements were assessed by Integrative Genomics Viewer (IGV)[77]. For RNA-seq data, gene fusions were detected using STAR-fusion (v1.4)[78].

### WES analysis
Whole-exome sequencing was performed using the Sure-Select Human All Exon V6 kit (Agilent, Santa Clara, CA) on tumor samples and matched NAT samples. Genomic alterations, including single base substitutions (single-nucleotide variants), short and long insertions/deletions (indels), copy number alterations were assessed.

### Somatic variant detection
Read-depth statistics were calculated using the DepthOfCoverage function in the Genome Analysis Toolkit (GATK v3.8.1.0)[79]. Paired-end reads in Fastq format were aligned to a reference human genome[80] (UCSC Genome Browser, hg19) using Burrows-Wheeler Aligner. Variant calling was conducted following GATK best practices. Somatic single-nucleotide variations and small insertions and deletions were detected using MuTect2 (GATK v4.1.2.0) and were annotated using ANNOVAR[81] based on UCSC known genes. The Maftools R package[82] was used to display mutant genes with non-synonymous mutations. MutSigCV[32] was used to identify significantly mutated genes with default parameters. Genes with $q < 0.01$ were identified as significantly mutated genes. Oncogenes and tumor suppressor genes were obtained from OncoKB (https://www.oncokb.org/)[83].

### Mutational signatures
SBSs are defined as a replacement of a certain nucleotide base. There are six possible substitutions: C > A, C > G, C > T, T > A, T > C, and T > G. Considering the nucleotide context, these SBS classes can be further expanded to 96 possible mutation types. The frequencies of the 96 mutation types were estimated for each sample. The non-negative matrix factorization algorithm of Sigminer[84] was used to exact mutational signatures of tumor samples. Signatures were compared against signatures derived from COSMIC (https://cancer.sanger.ac.uk/cosmic)[48] and cosine similarity was calculated to identify the best match.

### CNA calling
CNAs were called following somatic CNA best practice, using the CalculateTargetCoverage function in GATK (v4.1.2.0). We applied Genomic Identification of Significant Targets in Cancer (GISTIC2.0)[53] to identify significantly amplified or deleted focal-level and arm-level events, with $q < 0.05$ considered significant. The following parameters were used: amplification threshold = 0.1; deletion threshold = 0.1; cap value = 2.0; broad length cutoff = 0.90; remove X-chromosome = 0; confidence level = 0.95; join segment size = 4; arm-level peel off = 1; maximum sample segments = 2000; gene GISTIC = 1. Each gene in each sample is assigned a threshold copy number that reflects the magnitude of its deletion or amplification.

### Protein extraction and trypsin digestion
Samples were minced and lysed in lysis buffer (8 M urea, 100 mM Tris hydrochloride, pH 8.0) containing protease and phosphatase inhibitors (Thermo Scientific) and then sonicated for 1 min (3 s on and 3 s off, amplitude 25%). The lysates were centrifuged at $14,000 \times g$ for 10 min and supernatants were collected as whole-tissue extracts. Protein concentrations were determined by the Bradford protein assay (TaKaRa, T9310A). Extracts were reduced with 10 mM dithiothreitol at 56 °C for 30 min and alkylated with 10 mM iodoacetamide at room temperature in the dark for 30 min. The samples were digested with trypsin using a filter-aided sample preparation method[85].

### Peptide pre-fractionation
Tryptic peptides (50 µg) were separated in a home-made reverse-phase C18 column. Peptides were eluted and separated into nine fractions using an acetonitrile gradient (6%, 9%, 12%, 15%, 18%, 21%, 25%, 30%, and 35%) at pH 10. The nine fractions were pooled into three fractions (6% + 15% + 25%; 9% + 18% + 30%; 12% + 21% + 35%), vacuum-dried (Concentrator Plus, Eppendorf).

### Enrichment of phosphopeptides
Phosphopeptides were enriched by High-Select™ Fe-NTA Phospho-peptide Enrichment Kit (Thermo Fisher, A32992), according to the manufacturer's instruction. Briefly, 1 mg peptides were resuspended in 200 µL binding/wash buffer and loaded to the equilibrated spin

column with Fe-NTA resin. The samples were mixed with resin by gently tapping and then incubated for 30 min. The mixture was centrifuged at 1000×$g$ for 30 s to discard the flowthrough and then washed by 200 μL of binding/wash buffer for 3 times and washed by 200 μL of LC-MS grade water one additional time. The enriched phosphopeptides in NTA resin were eluted by adding 100 μL of elution buffer and centrifuged at 1000×$g$ for 30 s for two times and vacuum-dried.

## LC-MS/MS
Samples were analyzed on a Q Exactive HF-X mass spectrometer (Thermo Fisher Scientific) coupled with a high-performance liquid chromatograph (EASY-nLC 1200 System, Thermo Fisher Scientific). Dried peptide samples were dissolved in solvent A (0.1% formic acid in water) and loaded onto a trap column (100 μm × 2 cm, home-made; particle size, 3 μm; pore size, 120 Å; SunChrom) with a maximum pressure of 280 bar using solvent A, then separated on a home-made 150 μm × 12 cm silica microcolumn (particle size, 1.9 μm; pore size, 120 Å; SunChrom) with a gradient of 5–35% mobile phase B (acetonitrile and 0.1% formic acid) at a flow rate of 600 nL/min for 75 min. MS analysis was conducted with one full scan (300–1,400 m/z, $R$ = 120,000 at 200 m/z) at an automatic gain control target of 3e6 ions, followed by up to 20 data-dependent MS/MS scans with higher-energy collision dissociation (target 5e4 ions, max injection time 20 ms, isolation window 1.6 m/z, normalized collision energy of 27%). Detection was done using Orbitrap ($R$ = 7500 at 200 m/z). Data were acquired using the Xcalibur software (Thermo Fischer Scientific).

For the phosphoproteomic samples, enriched peptides were separated with 150 μm × 12 cm silica microcolumn in 150 min gradient (0–10 min, 3–8% of buffer B; 10–125 min, 8–16% of buffer B; 126–140 min, 25% of buffer B; 141–150 min, 95% of buffer B). MS analysis was conducted with one full scan (300–1400 m/z, $R$ = 120,000 at 200 m/z) at an automatic gain control target of 3e6 ions, followed by up to 20 data-dependent MS/MS scans with higher-energy collision dissociation (target 5e4 ions, max injection time 100 ms, isolation window 1.6 m/z, normalized collision energy of 27%). Detection was done using Orbitrap ($R$ = 15,000 at 200 m/z). Data were acquired using the Xcalibur software.

## Proteome identification and quantification
Raw files were processed in Firmiana[86] and searched against the human National Center for Biotechnology Information (NCBI) RefSeq protein database (updated on 04-07-2013, 32,015 entries) using the Mascot 2.4 search engine (Matrix Science Inc). Mass tolerances were 20 ppm for precursor and 50 mmu for product ions. Up to two missed cleavages were allowed. Cysteine carbamidomethylation was set as a fixed modification and methionine N-acetylation and oxidation as variable modifications. For phosphoproteomic samples, phosphorylation at Ser/Thr/Tyr was set as an additional variable modification. Precursor ion score charges were limited to +2, +3, and +4. The data were also searched against a decoy database so that protein identifications were accepted at an FDR of 1%. Label-free protein quantifications were calculated using a label-free, intensity-based absolute quantification (iBAQ) approach[87]. Match between runs[88] was used to improve parallelism between tumor/adjacent samples. We built a dynamic regression function based on common peptides in tumor/adjacent samples. Based on the correlation value $R^2$, Firmiana chooses a linear or quadratic function for regression to calculate the retention time (RT) of corresponding hidden peptides and checks the existence of the extracting ion current (XIC) based on the m/z and calculated RT. The program determines the peak area values of existing XICs. We calculated peak area values as parts of corresponding proteins. Proteins with at least 1 unique peptide with a 1% FDR at the peptide level were selected for further analysis. The FOT was used to represent the normalized abundance of a particular protein across samples. FOT was

defined as a protein's iBAQ divided by the total iBAQ of all proteins identified in a given sample. FOT values were multiplied by 10e10 for ease of presentation.

## MS platform quality control
For QC of MS performance, tryptic digests of HEK293T (ATCC: CRL-11268; RRID: CVCL_QW54) cell lysates were measured as a QC standard every 2 days for proteome analysis and a QC standard per day for phosphoproteome analysis. The HEK293T cell line, obtained from ATCC, was authenticated by short-tandem repeat profiling and was tested negative for mycoplasma contamination. The QC standard was made and run using the same method, conditions, software, and parameters as those used for tRCC samples. Pairwise Pearson's correlation coefficients were calculated and shown in Supplementary Fig. 1d–e.

## Preprocessing of proteomic data
The fraction of total (FOT) was used to represent the normalized abundance of a particular protein across samples. FOT was defined as a protein's iBAQ divided by the total iBAQ of all proteins identified in a given sample. FOT values were multiplied by 10e10 for ease of presentation and log2 transformed. The density plot of the normalized intensities of the proteins identified in each sample showed that all samples showed an expected unimodal distribution. K-nearest neighbor (KNN) imputation was applied to impute the missing values using R package DreamAI[89]. Proteins having more than 75% missing data were excluded to ensure that each sample had enough data for imputation (Supplementary Data 3).

## RNA-seq analysis
RNA-seq raw data quality was assessed with the FastQC (v0.11.9) and the adaptor was trimmed with Trim_Galore (v0.6.6) before any data filtering criteria was applied. Reads were mapped onto the human reference genome (GRCh38.p13 assembly) by using STAR software (v2.7.7.a). The mapped reads were assembled into transcripts or genes by using StringTie software (v2.1.4)[90] and the genome annotation file (GCF_000001405.39_GRCh38.p13_genomic.gff). For quantification purpose, the relative abundance of the transcript or gene was measured by a normalized metrics, FPKM (Fragments Per Kilobase of transcript per Million mapped reads). Transcripts with median FPKM > 1 were retained. Missing values were also imputed using KNN.

## Analysis of kidney nephron atlas expression data
An external expression dataset of normal tissue microdissected from various regions of the nephron[91] was used to infer the origin of renal malignancies (including chromophobe renal cell carcinoma and renal medullary carcinoma) in previous studies[35,36]. For each gene in the kidney cancer dataset (combined tRCC, and each 50 cases of ccRCC, PRCC, ChRCC from TCGA), we centered expression values on the mean centroid of these malignancies. The gene expression profiles from different nephron sites (both human and mouse) obtained from Cheval L. et al.[91] were centered on mean centroid across samples. We computed the global inter-profile correlation (by Pearson's), using all ~4000 genes in common, as previously described[35,36].

## Protein and pathway alterations in tumor vs. NATs
PCA was conducted to visualize the separation of tumor and tumor-adjacent proteomes using the R package factoextra v1.0.6. In total, proteins identified in both >25% of tumor and tumor-adjacent samples were used for subsequent analysis. Volcano plots were used to display DEPs in tumor and adjacent tissues by applying thresholds of fold-change >2 and Benjamini–Hochberg-adjusted $p < 0.05$. Among the DEPs, 891 proteins were significantly upregulated and 836 proteins were significantly downregulated in tRCC tumor tissues. The

DEPs were then subjected to functional enrichment analyses in ConsensusPathDB[92] (Supplementary Data 4).

## GSEA

GSEA was conducted using the GSEA 4.0.3 software (http://software.broadinstitute.org/gsea/index.jsp)[93]. KEGG, Wikipathways, and HALL-MARK gene sets downloaded from the MSigDB v7.1 (http://software.broadinstitute.org/gsea/msigdb/index.jsp) were set as background. $P$ value <0.05 was used as a cutoff. The normalized enrichment score was used to reflect the degree of pathway overrepresentation.

## Immune, stromal and pathway scores and TFE3/TFEB activities inferring

Immune, and stromal scores were inferred using the R package, ESTIMATE v1.0.11[94]. Gene sets were obtained from MSigDB v7.1 and the transcription factor (TFEB/TFE3) targets were collected from DoRothEA[37]. Transcription factor activity and pathway score for each sample was inferred using single-sample GSEA (ssGSEA)[38].

## Proteomic subtyping of tRCC

Consensus clustering was conducted using the R package CancerSubtypes[95] using Pearson correlation as the distance measure and the following detail settings were used for clustering: number of repetitions = 1000 bootstraps; item subsampling proportion = 0.8; feature subsampling proportion = 1). The 50% of the DEPs, between tumors and NATs, with the highest median absolute deviation in tumor samples were used for partitioning around medoids (PAM) clustering with up to six groups. Consensus matrices for $k = 2, 3, 4, 5, 6$ clusters are shown in Supplementary Fig. 6a. The consensus matrix for $k = 3$ showed clear separation among clusters. The cumulative distribution function of the consensus matrix for each k-value was also measured (Supplementary Fig. 6b). The relative change in area under the cumulative distribution function curve increased by 33% from 2 clusters to 3 clusters, whereas others exhibited no appreciable increase. Moreover, the average silhouette distance (0.75) for $k = 3$ was larger than $k = 4$ or $k = 5$ and did not have significant negative values. Based on the evidence above, the tRCC proteomic data were clustered into three groups (Supplementary Fig. 6c).

Subtype-specific upregulated proteins are: (1) detected in >25% tumor samples; (2) expressed higher than other subtypes (FC > 2, Wilcoxon rank-sum test, $p < 0.05$). Subtype-specific upregulated proteins were further analyzed in ConsensusPathDB[92]. DEPs of each subtype and relevant enriched pathways are listed in Supplementary Data 6.

## Immune subtype identification

To evaluate the tumor immune microenvironment of tRCC tumors, the raw enrichment scores of 64 different cell types were computed via xCell[52], based on the tumor proteomic profiles (Supplementary Data 7). Consensus clustering was performed using the R package ConsensusClusterPlus based on the $z$ score normalized Raw enrichment scores of tumor samples. Specifically, 80% of the original tRCC tumor samples were randomly subsampled without replacement and were partitioned into three major clusters using the PAM algorithm, which was repeated 2000 times.

## Correlations between subtypes and clinical features

To evaluate correlations between proteomic subtypes and clinical and genomic features, Fisher's exact test was conducted on categorical variables, including age, sex, TMB, CIN, mutational signatures, and significant arm-level CNA events. Only variables that varied significantly among the three proteome subtypes are shown in Fig. 6a.

## Effects of CNAs

Spearman's correlations between CNA values (gene level) and protein abundances were calculated using genes quantified at both CNA and proteome levels. CNAs with significant correlation with proteins were selected based on $p$ value <0.05 and used for further analysis. Gene-wise correlations with FDR < 0.1 were visualized using the R package multiOmicsViz. Genomic alterations that affect gene expression at the same locus are said to act in cis, whereas an impact of another locus is defined as a trans-effect, whereas the impact of other locus was defined as a trans-effect.

## Survival analysis

The Kaplan–Meier method was used for survival analyses, and groups were compared using the log-rank test. The R survival package 3.2–3 (Therneau and Lumley, 2015) and survminer 0.4.8 were used for statistical tests and visualization. The HR was calculated by Cox proportional hazards regression analysis. Variates with $p < 0.05$ were considered to significantly impact prognosis. CNA events with $p < 0.05$ in single variant analysis were selected for Cox regression multivariate analysis (Supplementary Data 2).

## IHC and immunofluorescence staining assays

Immunohistochemistry staining was conducted to assess the expression levels of DST and TBK1 using primary antibody against DST (#DF6752; Affinity Bioscience, China), TBK1 (#DF7026; Affinity Bioscience) at 1:3000 dilution, and peroxidase-conjugated goat anti-rat IgG (Abcam, #ab205718, 1:5000) according to manufacturer's protocols. The overall IHC score ranging from 0 to 12 was measured based on the multiplying of the staining intensity and extent score, as previously described[96].

During immunofluorescence assay, FFPE tissue slides were first deparaffinized in a BOND RX system (Leica Biosystems) and then incubated sequentially with primary antibodies targeting α-SMA (ServiceBio, #GB111364, 1:1000) and pan-CK (Abcam, #ab7753, 1:100). This was followed by incubation with secondary antibodies and corresponding reactive Opal fluorophores, and nuclei acids were stained with DAPI. Tissue slides that were bound with primary and secondary antibodies but not fluorophores were included as negative controls to assess autofluorescence. All scans for each slide were then superimposed to obtain a single image. Multilayer images were imported to inForm v.2.4.8 (Akoya Biosciences) for quantitative image analysis. The quantities of various cell populations were expressed as the number of stained cells per square millimeter and as the percentage of positively stained cells in all nucleated cells.

## Statistical analysis

Quantification methods and statistical analysis methods for proteomic and integrated analyses were mainly described and referenced in the respective Method Details subsections.

Additionally, standard statistical tests were used to analyze the clinical data, including but not limited to Wilcoxon rank-sum test, Fisher's exact test, Kruskal–Wallis test, and log-rank test. Statistical significance was considered when $p$ value <0.05. To account for multiple testing, the $p$ values were adjusted using the Benjamini–Hochberg FDR correction. Kaplan–Meier plots (log-rank test) were used to describe survival. Variables associated with overall survival were identified using univariate Cox proportional hazards regression models. Significant factors in univariate analysis were further subjected to a multivariate Cox regression analysis. All the analyses of clinical data were performed in R and GraphPad Prism.

## Reporting summary

Further information on research design is available in the Nature Portfolio Reporting Summary linked to this article.

## Data availability

Proteome and phosphoproteome raw datasets have been deposited to the ProteomeXchange Consortium (dataset identifier: PXD035377) via the iProX partner repository (https://www.iprox.cn/)[97] under Project ID: IPX0003336000. WES and RNA-seq data are available at NODE (The National Omics Data Encyclopedia) under Project ID: OEP002630. WES and RNA-seq data are also deposited in GSA-human (Genome Sequence Archive for human)[98] in NGDC (the National Genomics Data Center) (https://ngdc.cncb.ac.cn/)[99] under the accession: HRA003190 and HRA002855. The raw sequencing data are available under controlled access due to data privacy laws related to patient consent for data sharing and the data should be used for research purposes only. Access can be obtained by approval via their respective DAC (Data Access Committees) in the GSA-human database. According to the guidelines of GSA-human, all non-profit researchers are allowed access to the data and the Principle Investigator of any research group is allowed to apply for Controlled access of the data. The user can register and login to the GSA database website (https://ngdc.cncb.ac.cn/gsa-human/) and follow the guidance of "Request Data" to request the data step by step (https://ngdc.cncb.ac.cn/gsa-human/document/GSA-Human_Request_Guide_for_Users_us.pdf). The approximate response time for accession requests is about 2 weeks. The access authority can be obtained for Research Use Only. The user can also contact the corresponding author directly. Once access has been granted, the data will be available to download for 3 months. The remaining data are available within the Article, Supplementary Information, or Source Data file. Source data are provided with this paper.

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

## Acknowledgements

This work is supported by National Key R&D Program of China (2022YFA1303200 [C.D.], 2022YFA1303201 [C.D.], 2020YFE0201600 [C.D.], 2018YFE0201600 [C.D.], 2018YFA0507500 [C.D.], 2018YFA0507501 [C.D.], 2017YFA0505100 [C.D.], 2017YFC0908404 [C.D.], 2017YFA0505102 [C.D.], 2016YFA0502500 [C.D.], 2018YFE0201603 [C.D.], 2017YFA0505101 [C.D.], 2019YFC1316000 [H.L.Z.], 2019YFC1316005 [H.L.Z.]); National Natural Science Foundation of China (31770886 [C.D.], 31972933 [C.D.], 31700682 [C.D.], 82172817 [Y.Y.Q.], 82172741 [D.W.Y.], 32201215 [J.W.F.]); Shanghai Municipal Health Bureau Project (No. 2020CXJQ03 [D.W.Y.]); Natural Science Foundation of Shanghai (20ZR1413100 [H.L.Z.]); Shanghai Municipal Science and Technology Major Project (2017SHZDZX01 [C.D.]); Program of Shanghai Academic/Technology Research Leader (22XD1420100 [C.D.]); Shuguang Program of Shanghai Education Development Foundation and Shanghai Municipal Education Commission (19SG02 [C.D.]); Shanghai "Science and Technology Innovation Action Plan" medical innovation research Project (22Y11905100 [Y.Y.Q.]); Major Project of Special Development Funds of Zhangjiang National Independent innovation Demonstration Zone (ZJ2019-ZD-004 [C.D.]); Fudan original research personalized support project (C.D.); Shanghai Sailing Program (22YF1403100 [J.W.F.]); Beijing Xisike Clinical Oncology Research Foundation (No. Y-HR2020MS-0948), and Shanghai Anti-Cancer Association Eyas Project (No. SACA-CY21A06 [W.H.X.] and No. SACA-CY21B01 [A.H.T.M.J.A.W.E.]).

## Author contributions

Y.Y.Q., H.L.Z., J.Y.Z., D.W.Y., and C.D. conceived and planned the project. Y.Y.Q., A.A., W.H.X., Y.Z., X.T., J.Q.S., G.H.S., D.L.C., F.J.X., Y.W., H.L.G., S.J.N., and M.H.S. were responsible for sample and clinical information collection. X.H.W., X.R.P., L.B., and G.J.Y. contributed to sample preparation. Y.Y.Q., X.H.W., A.A., J.W.F., W.H.X., X.R.P., Y.Z., and Y.L. analyzed the data and contributed to the interpretation of the results. C.D. took the lead in writing the manuscript. All authors provided critical feedback and helped shape the research, analysis, and manuscript.

## Competing interests

The authors declare no competing interests.

## Additional information

**Supplementary information** The online version contains
supplementary material available at

Jian-Yuan Zhao, Hailiang Zhang, Dingwei Ye or Chen Ding.

**Peer review information** *Nature Communications* thanks Mukesh Kumar
and the other, anonymous, reviewer(s) for their contribution to the peer
review of this work. Peer reviewer reports are available.

