## [Peer Review File · Nature Communications]

Proteogenomic Characterization of MiT Family Translocation Renal Cell CarcinomaReviewers' Comments:

Reviewer #1:

Remarks to the Author:

Qu et al. report a proteogenomic characterisation of a rare subtype of kidney cancer, MiT family translocation renal cell carcinoma (tRCC), a tumour type that has been less well characterised than the other more common forms of RCC. For a rare tumour type the collection is substantial with 84 cases. The tumours have been characterised comprehensively, with all cases having undergone exome sequencing, the vast majority proteomic analysis, and smaller subsets transcriptomics and phosphoproteomics. Overall this represents a very interesting data sets and it will no doubt be useful to the field. The analyses are extensive and various subtype analyses and molecular correlations are presented with possible interpretations. I have the following specific comments.

1. The authors state in the introduction that tRCCs can present with various different histology, but I did not see information or discussion of the histopathological features of the tumours presented in this study. Histopathological annotation of the cases would significantly improve the study. Information on tumour grade should be minimally incorporated in figures 6A and 7A.
2. Line 125, the analysis of correlation between mRNA and protein levels. The number of proteins detected in tumours, ranging between 2,837 to 8,120, was substantially lower than the genes detected my mRNA analysis. Does this affect the correlation analysis?
3. In Fig. 7M-O, correlation between fibroblasts and an EMT signature is presented. Could this just reflect the fact that fibroblasts are mesenchymal? Is there any evidence that EMT in this analysis reflects the tumour cell state?
4. In several places the text implies causality when in fact the data is only correlative. For example, on line 227 the text states that "downregulation of most components of the cadherin-catenin complex increased the rate of progression or risk of death", line 326 "fusions displayed differential TFE3 activities, further influencing protein expression patterns and clinical outcomes", line 453 that "3p deletion can result in immune evasion and tumor metastasis", and line 408 that "14q deletions in tRCC resulted in poor prognosis by shaping the IM1-like microenvironment". The text should be carefully edited to make sure that such overstatements are corrected.

Minor

Abstract, "Thus, to the best of our knowledge, this study represents a comprehensive proteogenomic analysis of tRCC,...". While the statement hopefully is correct, I would consider rewording it slightly.

Line 163, "...indicating that impaired DDB1 was an important cause of SBS6 associated CIN and poor prognosis." needs revision. Do you mean reduced DDB1 levels or DDR?

Reviewer #2:

Remarks to the Author:

This is a comprehensive review of TFE3-translocated RCC from a multi-omics perspective. Some questions/suggestions are-

1. There are too few cases of TFE3 translocation RCC to draw any meaningful conclusions. An increase of such number of cases are warranted.
2. MITF RCC are now considered to include TFE3- translocated RCC, TFE3-translocated RCC and TFE3-amplified RCC. The latter i.e. TFE3-amplified RCC are now known to be aggressive with a particularly worse outcome and relatively specific morphologic features. TFE3-amplified RCC need to be incorporated in this cohort.
3. In absence of TFE3-translocated RCC and TFE3-amplified RCC, it would be advisable to present a

pure and exclusive cohort of TFE3-translocated RCC.

4. MITF RCC do not comprise of melanoma cells but may have cells which mimic such entities.

5. It is unclear how or why two fusion types are found in a single case (figure 1B)- does this case represent a collision tumor?

6. Can the authors provide some functional significance or background related to the most frequently mutated gene (FLG) in this cohort?

7. The female enrichment seen in this cohort needs to be clarified (In terms of sample/case collection bias).

8. Further data needs to be presented in terms of 3p deletion seen in a subset of cases and association with worse outcome- was canonical VHL loss seen in such cases? How were these cases excluded to be clear cell RCC?

9. It is surprising that TFE3 activity was seen to be similar amongst TFEB-tRCC and TFE3-trcc. Can the authors provide some rationale or speculate to this phenomenon?

10. It is again surprising that ASPSCR1 fusion is associated with a worse outcome (at a marginal p value). Are the LUC7L3 fusion cases associated with any distinctive clinical or morphologic features?

11. Do the GP1, GP2 and GP3 cases associate with different fusion partners in a significant fashion?

Are these cases associated with distinctive morphologic features which could be recognized clinically?

12. Are there any specific biomarkers associated with TFE3-TRCC which can help distinguish from other RCC like CCRCC, PRCC and CHRCC?

13. Can the authors provide some data to cell of origin for MITF-RCC and especially TFE3-RCC which are a subject of this study?

Reviewer #3:

Remarks to the Author:

The manuscript on tRCC represents an impressive amount of effort. Understandably the authors want to take credit for this effort and complete many analyses. While there are a few concerns about those analyses that I will point out below, the sheer number of analyses is overwhelming and will obscure the important results to the reader. To be acceptable for publication a revised manuscript would require significant shortening and movement of many figure elements to a supplement. The exact choices are up to the author, but the goal is to focus on the most important findings. Currently, the are listed in the abstract as:

- 1) Defective DNA repair played an important role in tRCC carcinogenesis and progression.
- 2) Metabolic processes were markedly dysregulated at both the mRNA and protein levels.
- 3) Proteomic data identified the mTOR signaling pathway as a potential therapeutic target and nominated several diagnostic biomarkers.
- 4) Moreover, we stratified tRCC into three subtypes (GP1-3): GP1, the most aggressive subtype, exhibited dominant complement cascades and GP3 showed rapid disease progression but had comparatively long overall survival.
- 5) Multi-omic integration revealed the dysregulation of cellular processes affected by genomic alterations, including oxidative phosphorylation, autophagy, transcription factor activity, and proteasome function.

It would appear to me that the most intriguing results would be (1), (3), and (4) and this is enough for an important paper. The data on #1 appear a bit weak so it would be important to clearly interpret the data supporting this position.

The major concerns regarding data analysis that need to be handled are:

- 1) It is inappropriate to list mutated genes not accounting for expected numbers of mutations. It is clear that this list is not a relevant tRCC gene list.
- 2) There is an overuse of marginal p values. There are dozens of tests presented in the paper, it is unclear how many other hypotheses were tested whose results were not included but given the very

large number of tests, the authors should use a q value or bon ferroni or just state what hypotheses they tested and then correct for them. Fundamentally, the plethora of p-values at 0.048 seem to be a consequence of over testing.

Reviewer #4:

Remarks to the Author:

The manuscript "Proteogenomic Characterization of MiT Family Translocation Renal Cell Carcinoma" by Qu et al., presents a proteogenomic analysis of microphthalmia transcription factor (MiT) family translocation renal cell carcinoma (tRCC) tumors and compare with normal adjacent tissues. The study may potentially serve as a resource for the community; however, the manuscript in its current form needs major revision before it is considered for publication in the journal Nature communication, which holds very high standard for publication.

Major Points:

1. I found paper to be too lengthy and at times it is not coherent. I would strongly recommend shortening the manuscript and focus on key and relevant discoveries.
2. In general, please expand the figure legend, if an abbreviation is used in the main body of figure, please write the full form in the figure legend. It is difficult to understand figure without proper legend, e.g., in Figure 2 A, 2B, it is not clear what KIRC, KIRP, KICH refers to and there are many such panels where details are missing.
3. Different number of patients were selected for different experiment, e.g., LFQ experiment were carried out on 74 tumors and 57 normal adjacent tissues (NATs), for phosphoproteomics 28 tumors and 21 paired NATs were used, for RNA sequencing 26 tumors and 16 paired NATs were used. It is not clear whether same patients and control were used for all the experiment or was there different set of patients for phosphoproteomics and RNA-Seq and there were some overlapping patients.
4. Is there a difference between "normal adjacent tissues" and "paired normal adjacent tissues", if so, please elaborate the difference and why different control subjects were chosen for different analysis?
5. Number of patients mentioned in text 28 tumors "line 87" does not match to number in the Figure 1A T (n=23) for phosphoproteomics.
6. In Figure 1F, both side of the Spearman's correlation it's written significant positive correlation, one should be negative correlation?
7. Number of proteins referred in Figure 1D, is it number of protein groups or individual protein? Also, the difference in protein identified between different samples ranges from 2833 to 8120 (line 119), I am wondering whether such a huge difference is because of technical or biological?
8. In line 120, a total of 33,676 phosphosites, corresponding to 6,469 phosphoproteins, were identified, are these unique phosphosites? Also, the number of identified phosphosites is highly variable, is it technical or biological?
9. What data was used for Figure 2A and 2B, and what does KIRC, KIRP and KICH represents?
10. In line 154, Figure 2D should be Figure 2E
11. In line 157, Figure 2E should be Figure 2F
12. In line 167, Figure 2I should be Figure 2J
13. In Figure 2L, what are GSR, GSS and GCLC?
14. In line 184, Figure 2P which does not exist should be Figure 2M
15. In line 179 author claim "SBS6 was positively correlated with the infiltration of CD8+ Tem, CD8+ Tgd, CD4+ naive, and CD4+ Tcm cells (Figure 2L)" however, Figure 2L does not represent this data and instead some other data which is not discussed in the manuscript.
16. Figure 3C and 3D are not readable
17. Figure 4I, please explain what data is used for panel A?
18. For comparison of TFEB-tRCC/TFE3-tRCC in Figure 5, because the number of samples are so different, TFEB (n=5) and TFE3 (n=63), it is very unbalanced number for statistical analysis and therefore this comparison should not be done especially given the patient heterogeneity and n=5 is a very small number. One way could be to perform analysis on same number (n=5) of randomly selected samples from both group and see how reproducible/similar the results are with current

analysis with very different number of samples for two group.

19. In Figure 7D, without axis and unit it is impossible to make any inference of the data.

20. In Figure 4D, both x-axis and y-axis reads Protein Log₂ FD (T/N), however, one axis should be mRNA, please check and correct.

21. Some of the panels can be moved to supplementary, e.g., I am not sure what is the relevance of Figure 5J ?

22. In Figure 6E, what is the color coding , what's the scale and what does the value represents?

Reviewer #1 (Remarks to the Author): expert in RCC genomics

Qu et al. report a proteogenomic characterisation of a rare subtype of kidney cancer, MiT family translocation renal cell carcinoma (tRCC), a tumour type that has been less well characterised than the other more common forms of RCC. For a rare tumour type the collection is substantial with 84 cases. The tumours have been characterised comprehensively, with all cases having undergone exome sequencing, the vast majority proteomic analysis, and smaller subsets transcriptomics and phosphoproteomics. Overall this represents a very interesting data sets and it will no doubt be useful to the field. The analyses are extensive and various subtype analyses and molecular correlations are presented with possible interpretations. I have the following specific comments.

Response: We appreciate the reviewers for the positive evaluation and constructive comments. We have revised the manuscript according to the comments. Especially, we updated the histopathological annotation of the tRCC cases. In addition, we carefully revised our text in a more appropriate and more accurate way.

Q1. The authors state in the introduction that tRCCs can present with various different histology, but I did not see information or discussion of the histopathological features of the tumours presented in this study. Histopathological annotation of the cases would significantly improve the study.

Information on tumour grade should be minimally incorporated in figures 6A and 7A.

Response: Many thanks for your valuable comments and pointing this out. In this study, we totally identified 15 different fusion types of *TFE3*-tRCCs and 4 fusion types of *TFEB*-tRCCs, and some of them presented different histopathological characteristics. It was noted that histopathological features varied even in the same fusion types. Here, we supplemented representative hematoxylin-eosin (H&E) staining to present various histopathological features of tRCC (magnification, 200×) (**Figure RL1**).

Figure RL1

A, *ASPCRI-TFE3* fusion tRCC. Tumor cells were papillary architecture, with medium-size, abundant cytoplasm, round and conspicuous nucleoli with significant atypia.

B, *SFPQ-TFE3* fusion tRCC. Tumor cells were glandular or nest-like distribution, and some of them were similar to clear cell renal cell carcinoma with apparent heteromorphism, inconsistent nuclei with obvious nucleoli, and interstitial lymphocyte infiltration.

C, *PRCC-TFE3* fusion tRCC. Tumor cells showed solid or nest-like distribution with abundant cytoplasm, medium to large nuclei with obvious atypia.

D, *NONO-TFE3* fusion tRCC. Tumor cells composed of carcinoma like clear cells showed papillary or nest-like distribution with small to medium in size, unclear boundaries, abundant cytoplasm and round nucleoli. Psammoma bodies were found in the local renal interstitials tissue indicated with the arrows).

E, *MED15-TFE3* fusion tRCC. Papillary architecture (right) merges with solid nested architecture (left).

F, *KHSRP-TFE3* fusion tRCC. Tumor structure showed solid with clear and abundant cytoplasm.

Psammoma bodies were found in the local renal interstitials tissue (indicated with the arrows).

G, *EIF4A2-TFEB* fusion tRCC. A solid nest pattern or papillary pattern resembling clear cell renal cell carcinoma.

H, *NEAT1-TFEB* fusion tRCC. Tumor cells were glandular or nest-like distribution with abundant, clear or eosinophilic cytoplasm, obvious nucleoli and significant atypia.

I, *ACTB-TFEB* fusion tRCC. Tumor cells were nest-like distribution with obvious nucleoli. Large amounts of melanin granules were observed in the cytoplasm of cells.

In addition, according to reviewer's suggestion, we updated the tumor stage and grade in the upper panels of **Figure 6A** and **7A**. The majority of tRCC cases showed ISUP grade 2 (n = 36, 41.9%) and grade 3 (n = 40, 46.5%), and the rest cases showed grade 4 (n = 10, 11.6%). We found that proteomic subtypes were significantly different in TNM stages (**Figure RL2A**). Specifically, GP1 had the highest proportion of stage III&IV cancers, while GP2 had the highest proportion of stage I&II cancers. In addition, we found GP1 cases contained more high-grade (grade 3&4) tumors, compared to GP2&3 cases (**Figure RL2B**). Immune subtypes showed no significant difference in TNM stage and ISUP grade.

Figure RL2

A, The association of proteomic/immune subtypes with TNM stage and ISUP grade (Fisher's exact test).

B, The proportion of different ISUP grades in GP1 and GP2&3 cases (One-sided Fisher's exact test).

Q2. Line 125, the analysis of correlation between mRNA and protein levels. The number of proteins detected in tumours, ranging between 2,837 to 8,120, was substantially lower than the genes detected my mRNA analysis. Does this affect the correlation analysis?

Response: We thank the reviewer for the comments. According to reviewer's comments, we further evaluated the correlation between transcriptome and proteome (**Figure RL3A**). The mRNA-protein correlation was moderate with sample-wise median Spearman's correlation of 0.39 (**Figure RL3A**), consistent with previous report (PMID: 32649875). In addition, we found that the identified protein numbers were not correlated with the mRNA-protein correlations (**Figure RL3B**). These results indicated that transcriptome and proteome were complementary for characterizing the different dimensions of molecular features of tumors. We have added these results in the **Supplementary Figure 1j**.

Figure RL3

A, Sample-wise mRNA-protein correlation of 20 tRCC samples.

B, Scatter plot showing the correlation between identified protein numbers and mRNA-protein correlations.

Q3. In Fig. 7M-O, correlation between fibroblasts and an EMT signature is presented. Could this just reflect the fact that fibroblasts are mesenchymal? Is there any evidence that EMT in this analysis reflects the tumour cell state?

Response: Thanks for the reviewer's comments. The reviewer is correct that our analysis could not distinguish the fact fibroblasts are mesenchymal or the EMT state of tumor cells. We conducted immunofluorescence (IF) analysis to elaborate the association between fibroblasts and EMT state of tumor cell. The results showed that IF co-staining of a large subset of tumors for pan-cytokeratin (CK), the epithelial marker, and alpha-smooth muscle actin (a-SMA), the fibroblast marker (**Figure RL4**), demonstrated the presence of tumor cells undergoing EMT and a small subset of fibroblast in IM1 tRCC tumors microenvironment. In addition, IM1 tRCC showed stronger co-staining of CK and a-SMA, and single staining of a-SMA than other subtypes of tRCC (**Figure RL4**), reflecting

the enhanced EMT and fibroblast infiltration in IM1 tumors. We have added these results in **Supplementary Figure 7e** in the revision.

Figure RL4

Representative immunofluorescence staining of α-smooth muscle actin (α-SMA) and pan-cytokeratin (CK) of a IM1 sample and a non-IM1 sample.

Q4. In several places the text implies causality when in fact the data is only correlative. For example, on line 227 the text states that "downregulation of most components of the cadherin-catenin complex increased the rate of progression or risk of death", line 326 "fusions displayed differential TFE3 activities, further influencing protein expression patterns and clinical outcomes", line 453 that "3p deletion can result in immune evasion and tumor metastasis", and line 408 that "14q deletions in tRCC resulted in poor prognosis by shaping the IM1-like microenvironment". The text should be carefully edited to make sure that such overstatements are corrected.

Response: Thanks for the considerate comments. According to reviewer's suggestion, we tuned down our statement to make the results more accurate. For example, "downregulation of most components of the cadherin-catenin complex increased the rate of progression or risk of death" was revised to "downregulation of most components of the cadherin-catenin complex was associated with poorer PFS"; "3p deletion can result in immune evasion and tumor metastasis" was revised to "3p deletion was correlated with the latent immune evasion and tumor metastasis"; "14q deletions in tRCC resulted in poor prognosis by shaping the IM1-like microenvironment" was revised to "14q deletion regulated the EMT and IM1-like microenvironment in tRCC, which was associated with poor prognosis". In brief, we had thoroughly revised manuscript to avoid overstatements.

Minor

Q5. Abstract, “Thus, to the best of our knowledge, this study represents a comprehensive proteogenomic analysis of tRCC,...”. While the statement hopefully is correct, I would consider rewording it slightly.

Response: Thanks for the comment. We revised our statement to “Thus, this study represents a comprehensive proteogenomic analysis of tRCC, providing valuable insights into its biological mechanisms, disease diagnosis, and prognostication.”

Q6. Line 163, "...indicating that impaired DDB1 was an important cause of SBS6 associated CIN and poor prognosis." needs revision. Do you mean reduced DDB1 levels or DDR?

Response: We thank the reviewer’s comment and apologize for the unclear description. We have revised this sentence as “Moreover, DDB1 abundance was negatively correlated with CIN and positively correlated with PFS, indicating that reduced DDB1 might be an important cause of SBS6 associated CIN.” in the revision.

Reviewer #2 (Remarks to the Author): expertise in translocated renal cell carcinoma

This is a comprehensive review of TFE3-translocated RCC from a multi-omics perspective. Some questions/suggestions are-

Q7. There are too few cases of TFE3 translocation RCC to draw any meaningful conclusions. An increase of such number of cases are warranted.

Response: We sincerely thank the reviewer for carefully reading our manuscript and the professional suggestions. Indeed, more *TFEB* translocation RCC cases analysis is required in order to draw significant conclusions. However, MiT family tRCC is a rare type of kidney cancer, among which *TFEB* translocation RCC even was a relatively rare type in the MiT family tRCC relative to *TFE3* translocation RCC. We screened 3,850 consecutive patients who underwent radical or partial nephrectomy for the treatment of renal tumors at Fudan University Shanghai Cancer Center (FUSCC, Shanghai, China), one of the largest cancer centers in East Asian, from January 2008 to December 2020, collecting 86 eligible tRCC cases. As the rarity of *TFEB*

translocation RCC, we collected 5 cases of *TFEB* translocated RCC, including 2 cases of *CLTC-TFEB* fusion, 1 case of *ACTB-TFEB* fusion, 1 case of *EIF4A2-TFEB* fusion and 1 case of *NEAT1-TFEB* fusion. The report of five *TFEB*-tRCC cases was relatively more than other reports. *TFEB* translocation RCC with different fusion partners, such as *KHDRBS2* inv(6)(p21; q11), *COL21A1* inv(6) (p21; p12), *CADM2* t(6; 3) (p21; p12), and *CLTC* t(6; 17) (p21; q23), have been reported in a single case (PMID: 24899691, 25401301, 31278395).

In addition, *TFEB*-tRCC and *TFE3*-tRCC samples were collected and processed with the same standard operating procedure, providing valuable multi-omic data. The comparison between *TFEB*-tRCC and *TFE3*-tRCC would improve our knowledge about different tRCC fusion types. We thank the reviewer again for the valuable comments, and will continue to collect more *TFEB* translocation RCC cases for further analyses in future.

Q8. MITF RCC are now considered to include TFE3- translocated RCC, TFEB-translocated RCC and TFEB-amplified RCC. The latter i.e. TFEB-amplified RCC are now known to be aggressive with a particularly worse outcome and relatively specific morphologic features. TFEB-amplified RCC need to be incorporated in this cohort.

Response: Thank you again for your positive comments and valuable advices to improve the quality of our research. More recently, RCC with *TFEB* amplification have been identified and appear to be associated with a more aggressive clinical course than *TFEB* translocated RCC. *TFEB* amplification in RCC can occur independently of or in association with *TFEB* rearrangement (PMID: 31382581, 31768316, 32251007). RCC with *TFEB* amplification was also discussed in the ISUP classification as an emerging renal cancer type (PMID: 32251007). We further surveyed *TFEB* rearrangement cases in our study, but unfortunately, we did not find any cases of *TFEB* amplified RCC. We will pay close attention to whether subsequent related cases have *TFEB* amplification, and update the relevant data and research progress in follow up studies.

Q9. In absence of TFEB-translocated RCC and TFEB-amplified RCC, it would be advisable to present a pure and exclusive cohort of TFE3-translocated RCC.

Response: Thanks for the comments. According to the reviewer's comments, we independently analyzed the proteomic data of TFE3-translocated RCC, including 69 tumors and 53 normal adjacent tissues (NATs).

As a result, we identified 1,647 proteins that showed significant differential expression (fold-change [FC] > 2; Benjamini-Hochberg adjusted $p < 0.05$), with 827 proteins downregulated and 820 upregulated in tumors compared to adjacent tissues (**Figure RL5A**). Compared with our overall cohort (*TFE3+TFEB* tRCC), a total of 920 upregulated proteins and 886 downregulated proteins were identified, of which 791 (85.98%) upregulated proteins and 777 (87.70%) downregulated proteins were identified in both cohorts (**Figure RL5B**). The differential analysis (including FC and p value) exhibited significant consistency between the *TFE3*-tRCC cohort and the overall cohort (**Figure RL5C**). Furthermore, enrichment analysis revealed that neutrophil degranulation, mTOR signaling, glycogen metabolism, insulin pathway, membrane trafficking and lysosome function were upregulated in tumors; while multiple metabolic pathway (valine, leucine and isoleucine degradation, biological oxidations, fatty acid metabolism and TCA cycle), proximal tubule bicarbonate reclamation and extracellular matrix organization were downregulated (**Figure RL5D**). The differential expressed proteins and corresponding enriched pathways in the *TFE3*-tRCC cohort showed a high degree of consistency with those in the overall cohort. In addition, we conducted proteomic subtyping and immune subtyping after *TFEB*-tRCC cases excluded, and found that proteomic subtypes and immune subtypes still showed significant correlation with patient clinical (**Figure RL5E**). The analysis results of *TFE3*-tRCC were added in the **Supplementary Data 4** in the revision.

Moreover, *TFEB*-tRCC and *TFE3*-tRCC samples were collected and processed with the same standard operating procedure, providing valuable multi-omic data. The comparison between *TFEB*-tRCC and *TFE3*-tRCC would improve our knowledge about different tRCC fusion types.

Figure RL5

A, Volcano plot showing DEPs (Benjamini–Hochberg adjusted p value < 0.05 , $FC > 2$) in tumor and normal adjacent tissues in the *TFE3*-tRCC cohort.

B, Overlap of DEPs in tumors and adjacent tissues (NATs) between the *TFE3*-tRCC cohort and the overall cohort. Red, up-regulated proteins; blue, down-regulated proteins.

C, Correlation of differential analysis between tumor and NATs in the *TFE3*-tRCC cohort (y axis) and the overall cohort (x axis). Upper, fold change; lower, BH adjusted p value.

D, DEPs in tumors and adjacent tissues, and their enriched biological pathways.

E, Kaplan–Meier curves of OS/PFS for proteomic subtypes and immune subtypes (log-rank test).

Q10. MITF RCC do not comprise of melanoma cells but may have cells which mimic such entities.

Response: Thank the reviewer for the comment. We apologized for the incorrect description. We agree with the reviewer’s point that MiTF tRCC do not comprise melanoma cell but comprise melanin pigment. We revised the sentence as “Some cases even comprise of perivascular epithelioid

neoplasm-like or melanoma-like differentiation cells”. Indeed, in our cohort, there was one case of *ACTB-TFEB* fusion tRCC (#tRCC_67) comprise of large amounts of melanin granules in the cytoplasm of tumor cells. Intraoperative specimen of this case appeared gray-black and coke-like to the naked eye. Postoperative pathological report showed that tumor cells were nest-like distribution with obvious nucleoli, and large amounts of melanin granules were observed in the cytoplasm of cells (**Figure RL6A**). After discussion at the Shanghai Clinicopathology Reading Conference, the possibility of MiT family translocation renal cell carcinoma was preliminarily considered, and *ACTB-TFEB* fusion tRCC was confirmed by genetic sequencing. The best characterized function of the MiT family is its master regulatory role in melanocyte differentiation (PMID: 25048860). It was reported that some tRCC contained melanin pigment (PMID: 34704642). We nominated 22 candidate tRCC biomarkers (**Figure RL6B-C**), among which GPNMB, GPR143, RAB9A, RAB32 were associated with melanosome biogenesis. GPNMB was a downstream target of MiTF family members. MiTF RCC showed increased MiTF transcriptional activities, thus upregulated expression of GPNMB, which were also reported in other studies (PMID: 21209915, 31043488).

Figure RL6

A, The H&E image of *ACTB-TFEB* fusion tRCC in this study. Large amounts of melanin granules were observed in the cytoplasm of cells.

B, The pipeline for tRCC biomarker identification.

C, Log 2-fold change between tumor and matched NATs is shown for the 22 tRCC biomarkers. These biomarkers are annotated with potential clinical utilities and IHC staining scores defined by

HPA.

Q11. It is unclear how or why two fusion types are found in a single case (figure 1B)- does this case represent a collision tumor?

Response: Thanks for the comments. Indeed, we reported two *TFE3*-tRCC cases, which separately harbored two different fusion types. The pathological features of these two cases by H&E staining indicated none of them were collision tumors (**Figure RL7A-B**). We confirmed that the tumor DNA was not contaminated during sequencing, and the somatic DNA matched the peripheral blood germ line. Based on the genetic test results, one of these patients harbored both *PTPN12-TFE3* and *RBM10-TFE3* fusions, while another patient co-harbored *SFPQ-TFE3* and *ZNF627-TFE3* fusions (**Figure RL7C-D**). We have marked these two cases in **Supplementary Data 1** in the revision.

Figure RL7

A, *PTPN12-TFE3* and *RBM10-TFE3* fusion tRCC. Tumor cells were nest-like or glandular distribution with intermediate in size, abundant cytoplasm and indistinct cell boundaries, round nuclei with inconspicuous nucleoli.

B, *SFPQ-TFE3* and *ZNF627-TFE3* fusion tRCC. Tumor cells were papillary or glandular distribution with intermediate in size, abundant cytoplasm, moderate amounts of clear cytoplasm, obvious nucleoli and severe cells atypia.

C-D, The IGV visualization of *PTPN12-TFE3* and *RBM10-TFE3* fusion tRCC and *SFPQ-TFE3* and *ZNF627-TFE3* fusion tRCC.

Q12. Can the authors provide some functional significance or background related to the most frequently mutated gene (FLG) in this cohort?

Response: Thank the reviewer for the comments. In the revision, MutsigCV was used to identify the significant mutated genes (SMGs) in the tRCC (**Figure RL8A-B**). We identified *BCDIN3D* (n = 6), *NDRG1* (n = 6), *ZNF668* (n = 6), and *GNPTG* (n = 4) as the SMGs in tRCC. However, the most frequently mutated genes, such as *FLG*, *TTN*, *MUC17*, et al. were not nominated as the SMGs, because those genes did not harbor more mutations than expected background mutation frequency (PMID: 23770567). *FLG*, *TTN*, *MUC17*, etc. were frequently mutated in other tumors (**Figure RL8C**), but not reported as the SMGs in those studies, probably due to these extremely long gene sequences (encoded protein > 4,000 amino acids) caused high-possibility of mutation (PMID: 23770567). Among the SMGs in tRCC, *NDRG1* acts as a tumor suppressor and plays an important role in p53-mediated caspase activation and apoptosis. The functions of *BCDIN3D*, *ZNF668*, and *GNPTG* in tumor are poorly studied. The functional significance of these SMGs in tRCC need to be further studied. We have updated this information in the revised manuscript.

Figure RL8

A, Genomic profile of 70 tRCC tumors with somatic mutations. Significant mutated genes were highlighted by stars.

B, The scatter plot showing the q value of MutsigCV and mutation frequencies of tRCC mutated genes.

C, The mutated frequencies of *FLG*, *TTN*, and *MUC17* in different tumor types (Data from <https://www.cbioportal.org/>).

Q13. The female enrichment seen in this cohort needs to be clarified (In terms of sample/case collection bias).

Response: Many thanks for careful reading our manuscript. The incidence of kidney cancer in male is higher than which in female, with a female: male ratio of 1.6: 1 (PMID: 31912902). However, tRCC is one of the fewer subtypes of RCC that occurs in female more frequently than male. In our study, the cohort consisted of 33.7% (29/86) male and 66.3% (57/86) female patients. An analysis

of 403 tRCC cases reported in the literatures also indicated that incidence of tRCC in female is higher than which in male (female: male ratio, 1.6:1) (PMID: 31382581). Zhuang *et al.* performed meta-analysis and find out the incidence rate of female patients is much higher than that of male patients with Xp11.2 tRCC. As Xp11.2 tRCC involved gene translocation and fusion in X chromosome and the number of X chromosomes in female is twice of male, it was possible to indicate that this particular incidence rate is related to the X chromosome (PMID: 32843027). Further research is required to discover the specific reasons for this. We have added the gender distribution of the tRCC in the **introduction** section of the revision.

Q14. Further data needs to be presented in terms of 3p deletion seen in a subset of cases and association with worse outcome- was canonical VHL loss seen in such cases? How were these cases excluded to be clear cell RCC?

Response: Thanks for your comment. The 3p deletion is a common event in ccRCC, as it leads to loss of the wild-type *VHL* allele and abrogation of VHL hypoxia-regulating activity. *VHL* mutations do not seem to play an important role in tRCC, as *VHL* mutations were not identified in our cohort. However, Gabriel *et al.* also identified the 3p deletion in 5 tRCC cases, and thus 3p deletion in tRCC tumors might be associated with inactivation of other tumor suppressor genes (PMID: 23817689). FISH assay is currently the gold standard for identification of *TFE3/TFEB* rearrangements (PMID: 31382581, 26880493). In our study, all cases were confirmed *TFE3/TFEB* rearrangements by FISH assays (**Figure RL8**), and fusion partners of *TFE3/TFEB* were identified by next-generation sequencing, which was described in the **Methods** in the revision.

Figure RL8

The representative FISH figures for *TFE3*-tRCC identification in our previous study (PMID: 26880493) (the same samples used in this study).

Q15. It is surprising that TFE3 activity was seen to be similar amongst TFEB-tRCC and TFE3-tRCC. Can the authors provide some rationale or speculate to this phenomenon?

Response: Thanks for reviewer's comments. We supposed that there might be several reasons for this phenomenon. First, TFEB was overexpressed in *TFEB*-tRCC than *TFE3*-tRCC. Accordingly, TFEB inferred activities were also stronger in *TFEB*-tRCC than in *TFE3*-tRCC (**Figure RL9A**). However, TFE3 protein levels were similar between *TFEB*-tRCC and *TFE3*-tRCC tumors. Consistently, TFEB activities were also similar amongst *TFEB*-tRCC than in *TFE3*-tRCC (**Figure RL9B**). Second, MiT transcription factors form homodimers or heterodimers that bind target promoters at a consensus E-box sequence motif (CA [C/T] GTG). Family members have overlapping transcriptional target specificity owing to the highly conserved bHLH-LZ domains (PMID: 25048860). Previous studies have indicated that two transcription factors regulate very similar sets of genes in multiple tissues and play a cooperative, rather than redundant role. Consistently, target genes of TFE3 and TFEB were significantly overlapped (**Figure RL9C**). These studies indicated that *TFEB* fusion could mimic the enhanced TFE3 activity caused by TFE3 fusion. Consistent with the idea of dosage, TFEB overexpression via viral-mediated gene transfer was able to rescue the phenotype of TFE3 KO mice and vice versa (PMID: 29764979).

Figure RL9

A, Comparisons of TFEB abundance and TFEB activities between *TFEB*-tRCC and *TFE3*-tRCC tumors (Wilcoxon rank-sum test).

B, Comparisons of TFE3 abundance and TFE3 activities between *TFEB*-tRCC and *TFE3*-tRCC tumors (Wilcoxon rank-sum test).

C, The overlap of TFE3 and TFEB target genes identified in our proteome data (hypergeometric test).

Q16. It is again surprising that *ASPSCR1* fusion is associated with a worse outcome (at a marginal p value). Are the *LUC7L3* fusion cases associated with any distinctive clinical or morphologic features?

Response: We sincerely thank the reviewer for your questions. We found that *ASPSCR1* and *LUC7L3* fusions were consistently more aggressive than other fusion types of *TFE3*-tRCC (**Figure RL10A**). Consistently, we found that *ASPSCR1* and *LUC7L3* fusion tRCC tumors have higher ISUP grade than other fusion types of *TFE3*-tRCC (**Figure RL10B**). More importantly, it was observed that *ASPSCR1* and *LUC7L3* fusion tRCC tumors showed higher TFE3 activities but lower kidney signature scores, compared with other *TFE3*-tRCC tumors (**Figure RL10C**).

To indicate the distinctive morphologic features of *LUC7L3-TFE3* fusion tRCC, we observed the pathological features of two *LUC7L3-TFE3* fusion tRCC cases by H&E staining. The common features of *LUC7L3-TFE3* fusion tRCC were abundant cytoplasm, indistinct cell boundaries, and obvious nucleoli in some cells (**Figure RL10D-E**).

Interestingly, *ASPSCR1* and *LUC7L3* were both located at 17q. Simultaneously, *ASPSCR1* and *LUC7L3 TFE3*-tRCC contained more 17q amplification events, which was associated with poorer PFS. The impacts of fusion types on CNA events need to be further studied.

Figure RL10

A, Kaplan–Meier curves of PFS for *ASPSCR1&LUC7L3* *TFE3*-tRCC tumors versus other *TFE3*-tRCC tumors.

B, Proportions of ISUP grades for *ASPSCR1&LUC7L3* *TFE3*-tRCC tumors versus other *TFE3*-tRCC tumors (One-sided Fisher’s exact test).

C, Comparison of TFE3 activities and kidney signature scores among different *TFE3*-tRCC fusion types (Wilcoxon rank-sum test).

D, Case 1. Tumor cells were solid and nest-like distribution with abundant cytoplasm and indistinct cell boundaries, severe cellular atypia, obvious nucleoli, and inclusion bodies (indicated with the arrow) in some cells.

E, Case 2. Tumor cells were glandular distribution with intermediate in size, abundant cytoplasm and indistinct cell boundaries, and obvious nucleoli in some cells.

Q17. Do the GP1, GP2 and GP3 cases associate with different fusion partners in a significant fashion?

Are these cases associated with distinctive morphologic features which could be recognized clinically?

Response: Thank the reviewer for the comments. We found that *TFEB*-tRCC was significantly enriched in GP3 tumors (**Figure RL11A**). In the revision, we updated histopathological annotation of the tRCC cases (**Figure RL11B**). The majority of tRCC cases showed ISUP grade 2

(n = 36, 41.9%) and grade 3 (n = 40, 46.5%), and the rest cases showed grade 4 (n = 10, 11.6%). We found GP1 cases contained more high-grade (grade 3&4) tumors, compared to GP2&3 cases (Figure RL11C). These results were added in the Figure 6A and Supplementary Figure 6D-E in the revision.

Figure RL11.

A, The proportion of *TFE3/TFEB*-tRCC in GP3 and GP1&2 cases (One-sided Fisher's exact test).

B, Distribution of fusion type, TNM stage, and ISUP grade among different tRCC proteomic subtypes.

C, The proportion of different ISUP grade in GP1 and GP2&3 cases (One-sided Fisher's exact test).

Q18. Are there any specific biomarkers associated with *TFE3*-tRCC which can help distinguish from other RCC like ccrcc, prcc and chrcc?

Response: Thanks for the comments. In the previous manuscript, we nominated 22 candidate tRCC biomarkers after the rigorous screening of proteomic data (Figure RL12A), to help the tRCC diagnosis. Among the 22 candidate biomarkers, ten showed low or no immunohistochemical (IHC) staining in > 90% of ccRCC. Among the ten proteins, glycoprotein nonmetastatic B (GPNMB) was identified as a diagnostic marker for *TFE3*-tRCC in a previous study (PMID: 31043488). We selected two candidates (DST, TBK1) from these candidate biomarkers and conducted IHC staining. The results showed that tRCC tumors presented stronger IHC staining than ccRCC, prcc, chrcc, and normal kidney tissue (Figure RL12B), indicating DST and TBK1 were potential biomarkers to distinguish tRCC and other kidney malignancies. We have added these results in the Supplementary Figure 2H in the revision. In addition, we are conducting multi-omic studies of other types of kidney malignancies. Our subsequent studies, comprehensive analysis of pan-kidney cancers, would further identify specific biomarkers to distinguish different types of RCC.

Response: We thank the reviewer for the valuable suggestion. An external expression dataset of normal tissue microdissected from various regions of the nephron (Cheval L, et al. PLoS One. 2012. PMID: 23056504) was used to infer the origin of renal malignancies (including chromophobe renal cell carcinoma and renal medullary carcinoma) in previous studies (Davis CF, et al. Cancer Cell. 2014. PMID: 25155756; Msaouel P, et al. Cancer Cell. 2020. PMID: 32359397). For each gene in the kidney cancer dataset (combined tRCC, and each 50 cases of ccRCC, PRCC, ChRCC from TCGA), we centered expression values on the mean centroid of these malignancies. The gene expression profiles from different nephron sites (both human and mouse) obtained from Cheval L, *et al* (Cheval L, et al. PLoS One. 2012. PMID: 23056504) were centered on mean centroid across samples. We computed the global inter-profile correlation (by Pearson's), using all ~4000 genes in common, as previously described (Davis CF, et al. Cancer Cell. 2014. PMID: 25155756; Msaouel P, et al. Cancer Cell. 2020. PMID: 32359397). The results showed that tRCC mRNA expression have a high degree of correlation with the proximal tubule (**Figure RL13**), indicating that similar to ccRCC and PRCC, proximal tubule was the origin of tRCC. We have added these results in the **Supplementary Figure 1K** in the revision.

Figure RL13

Heatmap showing inter-sample correlations (Pearson's r) between expression profiles of kidney malignancies (including tRCC, ccRCC, PRCC and ChRCC) and expression profiles of kidney nephron sites. S1 and S3, initial and terminal portions of the proximal tubule; mTAL, medullary thick ascending limb of Henle's loop; cTAL, cortical thick ascending limb of Henle's loop; DCT, distal convoluted tubule; CCD, cortical collecting duct; OMCD, outer medullary collecting duct.

Reviewer #3 (Remarks to the Author): expert in proteogenomics

The manuscript on tRCC represents an impressive amount of effort. Understandably the authors want to take credit for this effort and complete many analyses. While there are a few concerns about those analyses that I will point out below, the sheer number of analyses is overwhelming and will obscure the important results to the reader. To be acceptable for publication a revised manuscript would require significant shortening and movement of many figure elements to a supplement. The exact choices are up to the author, but the goal is to focus on the most important findings. Currently, the are listed in the abstract as:

- 1) Defective DNA repair played an important role in tRCC carcinogenesis and progression.
- 2) Metabolic processes were markedly dysregulated at both the mRNA and protein levels.
- 3) Proteomic data identified the mTOR signaling pathway as a potential therapeutic target and nominated several diagnostic biomarkers.
- 4) Moreover, we stratified tRCC into three subtypes (GP1–3): GP1, the most aggressive subtype, exhibited dominant complement cascades and GP3 showed rapid disease progression but had comparatively long overall survival.
- 5) Multi-omic integration revealed the dysregulation of cellular processes affected by genomic alterations, including oxidative phosphorylation, autophagy, transcription factor activity, and proteasome function.

Q20. It would appear to me that the most intriguing results would be (1), (3), and (4) and this is enough for an important paper. The data on #1 appear a bit weak so it would be important to clearly interpret the data supporting this position.

Response: Thank the reviewer for the cogitative comments. We apologize for the lack of highlights of the original manuscript.

According to reviewer's comments, we made a logical combing of our findings and adjusted the article structure of our manuscript to improve coherence.

First, the highlights could be summarized as four points: (1) Defective DNA repair plays an important role in tRCC carcinogenesis and progression. (2) Proteomic and phosphoproteome data identified the mTOR signaling pathway as a potential therapeutic target. (3) Molecular subtyping

and immune infiltration analysis reveal the inter-tumoral heterogeneity of tRCC. (4) Multi-omic integration revealed the dysregulation of cellular processes affected by genomic alterations, including oxidative phosphorylation, autophagy, transcription factor activity, and proteasome function.

Second, we moved the differential analysis of tumor and adjacent tissues to the second section in the **Results**. After the modification, the manuscript would be organized into sequential points: (1) differential analysis of tumor and adjacent tissues (**Figure 2** in the revision); (2) The impacts on clinical outcomes and molecular features of genomic alterations in tRCC (**Figure 3-5** in the revision). (3) Inter-tumoral heterogeneity of tRCC (**Figure 6-7** in the revision).

Third, we rearranged the findings in **Figure 5** and **Figure 7** to make our points clearer. In addition, according to the highlights above, we readjusted the supporting materials and removed redundant information. For example, **Figure 2H-I**, **Figure 3C-E**, et al. were moved to the Supplementary Figure in the revision. **Figure 2F**, **Figure 5E** et al. were removed from the manuscript.

Q21. The major concerns regarding data analysis that need to be handled are:

1) It is inappropriate to list mutated genes not accounting for expected numbers of mutations. It is clear that this list is not a relevant tRCC gene list.

Response: Thank the reviewer for the valuable comments. In the revision, MutsigCV was used to identify the significant mutated genes (SMGs) in the tRCC (**Figure RL14A-B**). We identified *BCDIN3D* (n = 6), *NDRG1* (n = 6), *ZNF668* (n = 6), and *GNPTG* (n = 4) were identified as the SMGs in tRCC. The most frequently mutated genes, such as *FLG*, *TTN*, *MUC17*, et al. were not nominated as the SMGs. It was because those genes did not harbor more mutations than expected background mutation frequency (PMID: 23770567). Among the SMGs in tRCC, *NDRG1* acts as a tumor suppressor and plays an important role in p53-mediated caspase activation and apoptosis. The functions of *BCDIN3D*, *ZNF668*, and *GNPTG* in tumor are poorly studied. We have updated this information in the revised manuscript. At present, the research of tRCC is still very limited, and the relevant mutated genes is unclear. In consequence, we listed the frequently mutated genes and significant mutated genes (highlighted by stars) in **Figure 1C** in the revision (**Figure RL14A**).

Figure RL14

A, Genomic profile of 70 tRCC tumors with somatic mutations. Significant mutated genes were highlighted by stars.

B, The scatter plot showing the q value of MutsigCV and mutation frequencies of tRCC mutated genes.

2) There is an overuse of marginal p values. There are dozens of tests presented in the paper, it is unclear how many other hypotheses were tested whose results were not included but given the very large number of tests, the authors should use a q value or bon ferroni or just state what hypotheses they tested and then correct for them. Fundamentally, the plethora of p-values at 0.048 seem to be a consequence of over testing.

Response: We thank the reviewer for the comments. The reviewer is correct that there are multiple marginal p values in the manuscript. (1-2) Gender and TKI response were associated with mutational signature SBS6 ($p = 0.049$, $p = 0.047$, respectively) (**Figure RL15A-B**). As the limitation of our sample sizes, these results needed further validation. We removed the relevant descriptions in the revision to improve the reliability of conclusions. (3) *ASPC1* and *LUC7L3* fusion tumors were more aggressive than other fusion type of tumors ($p = 0.048$) (**Figure RL15C**). *ASPC1* and *LUC7L3* fusion tRCC tumors have higher ISUP grade than other fusion types of *TFE3*-tRCC, which further supported the conclusion (**Figure RL15D**). The previous studies also reported that *ASPC1-TFE3* might be the most aggressive among the *TFE3* fusion genes (PMID: 21074195).

We supposed that the reason for the marginal p values was the limited sample size. The relatively

loose criteria were conducive to decrease the probability in making type II errors. We adjusted the p values when a large number of tests were used to avoid type I errors. For example, the p values derived from Wilcoxon rank-sum test, to identify the differentially expressed proteins among tRCC tumor tissues and normal adjacent tissues, were adjusted using the Benjamini–Hochberg FDR correction (**Methods**).

Figure RL15

A-B, Relationships between gender and SBS6 with TKI response in tRCC cohort (Fisher’s exact test). CR, complete response; PR, partial response; SD, stable disease; PD, progressive disease.

C, Kaplan–Meier curves of PFS for *ASPSCR1&LUC7L3* *TFE3*-tRCC tumors versus other *TFE3*-tRCC tumors.

D, Proportions of ISUP grades for *ASPSCR1&LUC7L3* *TFE3*-tRCC tumors versus other *TFE3*-tRCC tumors (One-sided Fisher’s exact test).

Reviewer #4 (Remarks to the Author): expertise in MS-based proteomics

The manuscript “Proteogenomic Characterization of MiT Family Translocation Renal Cell Carcinoma” by Qu et al., presents a proteogenomic analysis of microphthalmia transcription factor (MiT) family translocation renal cell carcinoma (tRCC) tumors and compare with normal adjacent tissues. The study may potentially serve as a resource for the community; however, the manuscript

in its current form needs major revision before it is considered for publication in the journal Nature communication, which holds very high standard for publication.

Response: We appreciate the reviewer for constructive comments and suggestions. We have revised the manuscript according to the comments. Especially, we adjusted the article structure of our manuscript to improve coherence. The usage of abbreviations and the order of figures were carefully checked in the revision. In addition, we enhanced the description of the quality assessment of our data.

Major Points:

Q22. I found paper to be too lengthy and at times it is not coherent. I would strongly recommend shortening the manuscript and focus on key and relevant discoveries.

Response: We appreciated the reviewer's profound suggestions. We apologize for the lengthiness and the lack of highlights in the original manuscript.

First, the highlights could be summarized as four points: (1) Defective DNA repair plays an important role in tRCC carcinogenesis and progression. (2) Proteomic and phosphoproteome data identified the mTOR signaling pathway as a potential therapeutic target. (3) Molecular subtyping and immune infiltration analysis reveal the inter-tumoral heterogeneity of tRCC. (4) Multi-omic integration revealed the dysregulation of cellular processes affected by genomic alterations, including oxidative phosphorylation, autophagy, transcription factor activity, and proteasome function. According to the highlights above, we rearranged the supporting materials and removed redundant information. For example, **Figure 2H-I, Figure 3C-E, et al.** were moved to the Supplementary Figure in the revision. **Figure 2F, Figure 5E et al.** were removed from the manuscript.

Second, we adjusted the structure of our manuscript to improve coherence. We moved the differential analysis of tumor and adjacent tissues to the second section in the **Results**. After the modification, the manuscript would be organized into sequential points: (1) differential analysis of tumor and adjacent tissues; (2) The impacts on clinical outcomes and molecular features of genomic alterations in tRCC. (3) Inter-tumoral heterogeneity of tRCC. In addition, we rearranged the findings in **Figure 5** and **Figure 7**.

In summary, we reorganized our story line and highlights, adjusted the structure of the article, and polished the language in revision. After revisions, we believed that the manuscript would be more concise and coherent.

Q23. In general, please expand the figure legend, if an abbreviation is used in the main body of figure, please write the full form in the figure legend. It is difficult to understand figure without proper legend, e.g., in Figure 2 A, 2B, it is not clear what KIRC, KIRP, KICH refers to and there are many such panels where details are missing.

Response: Thank for review's thoughtful suggestion. KIRC, KIRP, and KICH refers to clear cell renal cell carcinoma (KIRC), papillary renal cell carcinoma (KIRP), and chromophobe renal cell carcinoma (KICH), respectively. We examined all abbreviations used in our manuscript. We provided all abbreviations and corresponding full names in the **Figure legends** and **Supplementary Table 1** the revision.

Q24. Different number of patients were selected for different experiment, e.g., LFQ experiment were carried out on 74 tumors and 57 normal adjacent tissues (NATs), for phosphoproteomics 28 tumors and 21 paired NATs were used, for RNA sequencing 26 tumors and 16 paired NATs were used. It is not clear whether same patients and control were used for all the experiment or was there different set of patients for phosphoproteomics and RNA-Seq and there were some overlapping patients.

Response: Thanks for the comments. All omics experiments including WES, transcriptome, proteome, phosphoproteome were conducted on the same patients and control. As the rareness of tRCC and limited sample volumes, priority went to the meet the requirements of the detection of WES and proteome. When the sample was still sufficient after WES and proteome analysis, the remaining sample was used to phosphoproteomics and RNA-Seq analysis. As a result, WES was conducted on 84 paired tRCC samples. Samples from 2 patients were excluded due to low DNA quality. Label-free proteomic and phosphoproteomic approaches were carried out on 74 tumors and 57 NATs, and 28 tumors and 21 NATs, respectively. RNA-seq was carried out on 26 tumors and 16 NATs. As long as the volume of sample was sufficient, four dimensions of omics were analyzed. The **Figure RL16 (Figure 1A** in the manuscript) showed the patients used for different dimensions

of omics analysis.

Figure RL16

Schematic representation of tRCC multiomics analyses, including WES, RNA-seq, proteomics and phosphoproteomics.

Q25. Is there a difference between “normal adjacent tissues” and “paired normal adjacent tissues”, if so, please elaborate the difference and why different control subjects were chosen for different analysis?

Response: We thank the reviewer’s comments. In the original manuscript, we used “normal adjacent tissues” as control to reveal the molecular alterations of tRCC compared to normal adjacent tissues (NATs). Paired normal adjacent tissues was used to screen out potential biomarkers which showed higher expression levels in tumor tissues than paired normal adjacent tissues in more than 80% of tumor-NAT pairs.

According to reviewer’s suggestion, we evaluated the differences of tumor/normal adjacent tissues using “normal adjacent tissues” and “paired normal adjacent tissues” as control respectively (**Figure RL17A**). The results showed using paired normal adjacent tissues as control identified 960 tumor upregulated proteins and 837 tumor downregulated proteins. About 90% differential expressed proteins (DEPs) were the same as the DEPs identified using normal adjacent tissues as control (**Figure RL17A**). The enriched pathways were also consistent with the pathways enriched by the identified DEPs using normal adjacent tissues as control (**Figure RL17B-C**). The results of differential analysis between tumor and NAT using paired tumor and NAT samples were added in the **Supplementary Data 4**.

Figure RL17

A, Venn plots showing the identified differential expressed proteins by using normal adjacent tissues and paired normal adjacent tissues as control respectively.

B, Pathways enriched by differential expressed proteins, which were identified by using paired tumor and NATs.

C, Pathways enriched by differential expressed proteins, which were identified by using all proteome data.

Q26. Number of patients mentioned in text 28 tumors “line 87” does not match to number in the Figure 1A T (n=23) for phosphoproteomics.

Response: Thanks for review’s comment. We apologized for the mistake in **Figure 1A**. The number mentioned in the text is correct. We revised **Figure 1A** in the revision (**Figure RL18**).

Figure RL18

Schematic representation of tRCC multiomics analyses, including WES, RNA-seq, proteomics and phosphoproteomics.

Q27. In Figure 1F, both side of the Spearman's correlation it's written significant positive correlation, one should be negative correlation?

Response: Thank for review's comment. The review is correct that genes with positive mRNA-protein correlation were enriched in kidney elevated proteins (kidney signature), Gly/Ser/Thr metabolism, and extracellular matrix (ECM) receptor interaction, whereas genes with negative correlation were enriched in proteasome and oxidative phosphorylation (OXPHOS). We thank the review again for the comment.

Q28. Number of proteins referred in Figure 1D, is it number of protein groups or individual protein? Also, the difference in protein identified between different samples ranges from 2833 to 8120 (line 119), I am wondering whether such a huge difference is because of technical or biological?

Response: Thanks for the comments. We apologize for not explaining it clearly and thank the reviewer for pointing out. First, number of proteins referred in **Figure 1D** is number of protein groups. For quality control (QC) of the mass spectrometry performance, tryptic digests of HEK293T cell lysates were measured as a QC standard every 2 days. The QC standard was made and run using the same method, conditions, software, and parameters as those used for tRCC samples. The result showed that our mass spectrometry platform was robust (**Figure RL19A**).

Figure RL19B showed the distribution of the numbers identified proteins, with the average

identified protein number of 5,607. There were seven (5.34%) samples had identified protein numbers below 4,000 (**Figure RL19B**). We surveyed the total ion current (TIC) of MS1 and MS2 of all proteome samples. We found that samples with identified proteins less than 4,000 showed similar distribution of these parameters with other samples (**Figure RL19C**). In addition, we analyzed the correlation protein quantification results of the samples with protein identification numbers. We found that high-abundance proteins, especially hemoglobin, were negatively correlated with protein identification numbers, indicating samples with lower protein identification numbers contained large amounts of high-abundance proteins (**Figure RL19D**). Therefore, the difference in protein identified reflected the feature of label-free proteome analysis and the heterogeneity of samples. It might be caused by biological factors, rather than technical factors.

Figure RL19

- A**, Correlations of quality control sample quantification.
- B**, Distribution of protein numbers in tumors and NATs by a density plot.
- C**, The boxplot showing the TIC of MS1/MS2 of tRCC samples. Samples with identified protein less than 4,000 were noted (Wilcoxon rank-sum test).
- D**, Scatter plots showing the correlation between protein identification numbers and high-abundance protein abundances.

Q29. In line 120, a total of 33,676 phosphosites, corresponding to 6,469 phosphoproteins, were

identified, are these unique phosphosites? Also, the number of identified phosphosites is highly variable, is it technical or biological?

Response: Thank the reviewer for the comments. (1) These phosphosites are unique phosphosites. (2) The quality control of phosphoproteome data showed the high correlation of standard samples, with the median Pearson's correlation of 0.96 (**Figure RL20**), showing the stability of our mass spectrometry platform for phosphoproteome analysis. The numbers of identified phosphosites of standard samples were from 10,136 to 11,976, with the coefficient of variation (CV) of 0.04. We calculated the CV of the identified phosphosite numbers in our study (CV = 0.46) and a recent lung adenocarcinoma multiomics study (Xu JY, et al. Cell. 2020. PMID: 32649877) (CV = 0.36). These results suggested that the heterogeneity of tumor samples might cause the variable phosphoproteome identification. Thus, we supposed that the biological factors resulted the highly variability of identified phosphosites. We have added the quality control of phosphoproteome in the revision.

Figure RL20

Correlations of quality control sample quantification in phosphoproteome analysis.

Q30. What data was used for Figure 2A and 2B, and what does KIRC, KIRP and KICH represents?

Response: We thank the reviewer for the comment. KIRC, KIRP and KICH are abbreviations of clear cell renal cell carcinoma (KIRC), papillary renal cell carcinoma (KIRP), and chromophobe renal cell carcinoma (KICH), respectively. We added the full names of all abbreviations in the figure legend in the revision.

Q31. In line 154, Figure 2D should be Figure 2E

Response: We appreciate the reviewer for the comments. The reviewer is correct that we made

some mistakes in figure serial numbers. We carefully checked the manuscript and figures and revised it in the revision.

Q32. In line 157, Figure 2E should be Figure 2F

Response: We thanks for the comment. We have revised it.

Q33. In line 167, Figure 2I should be Figure 2J

Response: We thanks for the comment. We have revised it.

Q34. In Figure 2L, what are GSR, GSS and GCLC?

Response: We thank the reviewer for the comment. GSR, GSS and GCLC are glutathione (GSH) synthesis-related enzymes. GSR is Glutathione-Disulfide Reductase, GSS is Glutathione Synthetase, and GCLC is Glutamate-Cysteine Ligase Catalytic Subunit. We added the full names of these proteins in the figure legend in the revision. In addition, we examined all abbreviation used in the manuscript and confirmed that all the full names were given in the revision.

Q35. In line 184, Figure 2P which does not exist should be Figure 2M

Response: We thanks for the comment. We have revised it.

Q36. In line 179 author claim “SBS6 was positively correlated with the infiltration of CD8+ Tem, CD8+ Tgd, CD4+ naive, and CD4+ Tcm cells (Figure 2L)” however, Figure 2L does not represent this data and instead some other data which is not discussed in the manuscript.

Response: Thank the reviewer for the comments. We apologized for the mistake. This figure was moved to Supplementary figure which made our manuscript and figures not corresponding exactly. We had revised it and checked all of the figure orders in the revision.

Q37. Figure 3C and 3D are not readable

Response: Thanks for reviewer’s comment. Figure 3C-D showed the effects of copy-number alternations on mRNA and protein abundance. Significant positive and negative correlations (Spearman’s correlation, FDR < 0.10) are indicated by red and green. Genomic alterations that affect

gene expression at the same locus are said to act in *cis* (diagonal patterns), whereas an impact of another locus is defined as a *trans*-effect (vertical patterns). Our data showed that there are more *cis*-effects at mRNA level and more *trans*-effects at proteome level. We have revised the legend to make it legible. In addition, these figures were moved to **Supplementary Figure 4** in the revision.

Q38. Figure 4I, please explain what data is used for panel A?

Response: We thanks for the comment. To identify the potential biomarkers of tRCC, only proteins showing increased expressions in tumor in more than 80% tumor-normal adjacent tissue (NAT) pairs were screened out (**Figure RL21**, **Figure 4H** in the previous manuscript). To depict the distributions of pairwise tumor-NAT differences for the 22 biomarkers, log 2-fold change between tumor and matched NATs was shown in **Figure 4I** (**Figure 2I** in the revision).

Figure RL21

The pipeline for tRCC biomarker identification.

Q39. For comparison of TFEB-tRCC/TFE3-tRCC in Figure 5, because the number of samples are so different, TFEB (n=5) and TFE3 (n=63), it is very unbalanced number for statistical analysis and therefore this comparison should not be done especially given the patient heterogeneity and n=5 is a very small number. One way could be to perform analysis on same number (n=5) of randomly selected samples from both group and see how reproducible/similar the results are with current analysis with very different number of samples for two group.

Response: Thanks for review's suggestion. According to reviewer's suggestion, we used Monte Carlo permutation test to reanalyze our data. The results showed that p values from Wilcoxon rank-sum test were similar with which from Monte Carlo permutation test (**Figure RL22A**). The differential expressed proteins (DEPs) identified by permutation test covered more than 90% of the

DEPs identified by Wilcoxon rank-sum test (**Figure RL22B**). In addition, the overrepresented pathways using DEPs from Wilcoxon rank-sum test and DEPs from Monte Carlo permutation test were consistent (**Figure RL22C-D**). Consequently, the unbalanced number of *TFEB*-tRCC and *TFE3*-tRCC did not impact the analysis results.

Figure RL22

A, Scatterplot depicting p values from Wilcoxon rank-sum test and Monte Carlo permutation test comparing *TFE3*-tRCC with *TFEB*-tRCC (Spearman's correlation).

B, Venn plots showing the identified differential expressed proteins of *TFE3*-tRCC and *TFEB*-tRCC by using Wilcoxon rank-sum test and Monte Carlo permutation test respectively.

C, Pathways enriched by differential expressed proteins of *TFE3*-tRCC and *TFEB*-tRCC by using Monte Carlo permutation test.

D, Pathways enriched by differential expressed proteins of *TFE3*-tRCC and *TFEB*-tRCC by using Wilcoxon rank-sum test.

Q40. In Figure 7D, without axis and unit it is impossible to make any inference of the data.

Response: Thanks for review's comment. According to reviewer's suggestion, we redraw this figure (**Figure RL23**).

Figure RL23

Pathway scores of Complement Cascade and Epithelial mesenchymal transition among three immune subtypes (Kruskal–Wallis test).

Q41. In Figure 4D, both x-axis and y-axis reads Protein Log2 FD (T/N), however, one axis should be mRNA, please check and correct.

Response: We thanks for reviewer’s comment. The reviewer is correct that the y-axis should be mRNA log2 FC(T/N). We corrected this mistake in the revision (**Figure RL24**).

Figure RL24

Scatterplot depicting tumor-NAT differential expressions of at mRNA (y axis) and protein (x axis) levels.

Q42. Some of the panels can be moved to supplementary, e.g., I am not sure what is the relevance of Figure 5J ?

Response: Thanks for reviewer’s comment. Figure 5J showed the elevated TFE3 target gene encoded proteins in *ASPSCR1* and *LUC7L3 TFE3*-tRCC tumors. According to reviewer’s comments

in Q23, we adjusted the structure of the manuscript, and this figure was removed in the revision.

Q43. In Figure 6E, what is the color coding, what's the scale and what does the value represents?

Response: Thanks for the comments. We apologize for not explaining them clearly and thank the reviewer for pointing out. In the revision, we redraw **Figure 6E** and removed this figure to **Supplementary Figure 6**.

This figure exhibited proteins in four pathways that were differentially expressed in the three proteomic subtypes. The color coding represents the different abundance of protein expression of different subtypes. The scale represents the z-score of the mean protein expression of each subtype. Our data showed that proteins upregulated in GP1 were enriched in complement and coagulation cascades pathway; GP2 was more associated with oxidative phosphorylation; GP3 had upregulated proteins that were enriched in pathways related to DNA damage repair and translation. The values in the original version represent the ratio of protein expression in GP2 and GP3 to GP1, and we moved this figure to **Supplementary Figure 6f** in the revision.

Figure RL25

Proteins in four pathways (complement and coagulation cascades, oxidative phosphorylation, DNA damage repair and translation) that were differentially expressed in the three proteomic subtypes.

** See Nature Research's author and referees' website at www.nature.com/authors for information about policies, services and author benefits.

Reviewers' Comments:

Reviewer #1:

Remarks to the Author:

The authors have addressed my previous points. I have no further comments.

Reviewer #3:

Remarks to the Author:

1) It is inappropriate to list mutated genes not accounting for expected numbers of mutations. It is clear that this list is not a relevant tRCC gene list. This has not been fixed. TTN is not a relevant gene and inserting this into a manuscript is failing to recognize what we know. Failure to accept this criticism is deeply concerning.

2) There is an overuse of marginal p values. There are dozens of tests presented in the paper, it is unclear how many other hypotheses were tested whose results were not included but given the very large number of tests, the authors should use a q value or bon ferroni or just state what hypotheses they tested and then correct for them. Fundamentally, the plethora of p-values at 0.048 seem to be a consequence of over testing.

3) There are still 68 panels. This is unnecessary. It makes the paper hard to read.

Reviewer #4:

Remarks to the Author:

Authors have addressed all my concerns and I believe the manuscript is suitable for publication in journal nature communications.

Reviewer #5:

Remarks to the Author:

The authors have addressed the comments of Reviewer 2 in a satisfactory manner and no further edits are required.

Reviewer #3 (Remarks to the Author):

1) It is inappropriate to list mutated genes not accounting for expected numbers of mutations. It is clear that this list is not a relevant tRCC gene list. This has not been fixed. *TTN* is not a relevant gene and inserting this into a manuscript is failing to recognize what we know. Failure to accept this criticism is deeply concerning.

Response: Thanks for the comment. In the previous response, we noted that “The high frequently mutated genes, such as *FLG*, *TTN* etc., were not nominated as the SMGs, probably due to these extremely long gene sequences (encoded protein > 4,000 amino acids) caused high-possibility of mutation”. According to reviewer’s suggestion, we revised **Fig.1c (Figure RL1)** and removed the description of irrelevant mutant genes (*FLG*, *TTN* etc.) to avoid misleading.

Figure RL1

Genomic profile of 70 tRCC tumors with somatic mutations. SMGs, TSGs, and oncogenes are noted by different shapes.

2) There is an overuse of marginal p values. There are dozens of tests presented in the paper, it is unclear how many other hypotheses were tested whose results were not included but given the very large number of tests, the authors should use a q value or bon ferroni or just state what hypotheses they tested and then correct for them. Fundamentally, the plethora of p-values at 0.048 seem to be a consequence of over testing.

Response: We thank the reviewer for the thoughtful and helpful comments. After revision, there was one marginal p value used in the manuscript (**Figure RL2a**). We agreed with the reviewer that the plethora of p-values at 0.048 might be a consequence of over testing, which would cause a false positive. To reduce the probability of a false positive, we reviewed the pathological feature of

ASPSCR1 and *LUC7L3* fusion tRCC tumors. We found that *ASPSCR1* and *LUC7L3* fusion tRCC tumors have higher ISUP grade than other fusion types of *TFE3*-tRCC (**Figure RL2b**), which further supported the conclusion, *ASPSCR1* and *LUC7L3* fusion tumors were more aggressive than other fusion type of *TFE3*-tRCC tumors. In addition, a previous study reported that *ASPSCR1*-*TFE3* might be the most aggressive among the *TFE3* fusion genes (PMID: 21074195), supporting our conclusion.

In our previous response, we noted that a possible reason for the marginal p values was the limited sample size. It was reported that some cases of tRCC with histopathologic features indistinguishable from clear cell renal cell carcinoma (ccRCC) and papillary renal cell carcinoma (pRCC) were included in TCGA (PMID: 34986355). We incorporated data on these tRCC cases (n = 15) from the TCGA studies (PMID: 29617662) with our data, and reevaluated the association of *ASPSCR1* and *LUC7L3* fusion with patient clinical outcomes. Consistently, it showed *ASPSCR1* and *LUC7L3* fusion tumors had poorer survival than other fusion type of *TFE3*-tRCC, with a more significant difference (log-rank test, p = 0.033) (**Figure RL2c**). In summary, these results indicated our conclusion was reliable.

Figure RL2

a, Kaplan–Meier curves of PFS for *ASPSCR1*&*LUC7L3* *TFE3*-tRCC tumors versus other *TFE3*-tRCC tumors in this cohort (log-rank test).

b, Proportions of ISUP grades for *ASPSCR1*&*LUC7L3* *TFE3*-tRCC tumors versus other *TFE3*-tRCC tumors (One-sided Fisher’s exact test).

c, Kaplan–Meier curves of PFS for *ASPSCR1*&*LUC7L3* *TFE3*-tRCC tumors versus other *TFE3*-tRCC tumors combining this cohort and the TCGA cohort (log-rank test).

3) There are still 68 panels. This is unnecessary. It makes the paper hard to read.

Response: We appreciate the reviewer for comment. In our initial manuscript, there are 84 panels. After revision, the panel number was decreased to 68. According to reviewer's suggestion, we further reduced the panels to increase the readability of our manuscript in this revision. Specifically, **Fig. 3a-b, Fig. 4d-f, Fig. 6d-f, and Fig. 7e, f, j, k** were moved to the Supplementary Figures. After revision, there are 57 panels in the manuscript. This optimization would make the paper more concise and remain the conclusions unchanged. We thank the reviewer again for the helpful suggestion.

Reviewers' Comments:

Reviewer #3:

None